
# **Identifying Vulnerable Population in the Urban**
# **Society: a Case Study of Wuhan, China**
Jia Xu[1], Makoto Takahashi[2] and Weifu Li [3]
1 Department of Public Administration, Faculty of Humanities and Social Sciences, Dalian University
of Technology, Dalian 116081, China.
2 Department of Social and Human Environment, Graduate School of Environmental Studies, Nagoya
University, Nagoya 464-8601, Japan.
3 College of Science, Huazhong Agricultural University, Wuhan 430070, China.
*Correspondence to*: Jia Xu (xujia_ouc@163.com)
**Abstract.** In the context of unprecedented extreme weather and climate events, the internal structural
factors of society play a decisive role in the extent to which human beings are affected by disasters and
their ability to respond to disasters. In the past few decades, the rapid urbanization process in
developing countries represented by China has also greatly increased social vulnerability. The process
has generated uneven living conditions and created many vulnerable groups, including urban poverty,
migrants, and socially and geographically marginalized groups, who face difficulties in living
conditions, education, livelihood stability, and so on.
This study sets up indicators from a micro perspective: three indicators of exposure, four of sensitivity,
and eight of adaptive capacity are involved. Based on this evaluation index system, this study conducts
a social vulnerability assessment of the populations in Hongshan District, Wuhan City, China through
individual questionnaire surveys. *K*-means cluster analysis was used to get the high, medium, and low
levels of social vulnerability, which has achieved the comparison of different community types and the
identification of vulnerable groups.
The results show the close interrelationships between different types of communities in terms of
physical and built environments, and different levels of social vulnerability to disasters, in particular
pointing to the massive cluster of rural-to-urban migrants living in inferior urban villages, informal
settlements in the city, and suffering especially from instability of livelihoods. The quantitative
understanding of the dissimilarity in the degree of social vulnerability between different communities
and populations is of great significance for the reduction of social vulnerability and disaster risk
specifically and pointedly.

**Keywords:** Social vulnerability; Vulnerability index; *K*-means cluster analysis; Vulnerable groups;
Urban mosaic







**1 Introduction**

**1.1 Urbanization, Disaster risks and Social Vulnerability**

Extreme weather events have increased in frequency as a result of climate change in recent years. Especially, warming has become a predominant feature of the Earth's climate system, which has brought about changes in precipitation patterns and led to extreme weather such as heatwaves, droughts, forest fires, heavy rains, and floods. China is currently one of the most disaster-plagued countries in the world. There are many different kinds of disasters, and in recent years, their frequency, intensity, spatial scope, and duration has further expanded. Furthermore, the research indicated that extreme weather events would happen more frequently both in the world and in China, and human society's vulnerability would consequently grow more acute in the future (IPCC 2012, Huang et al. 2020).

Vulnerability is a key concept for both disaster risk and climate change adaptation. By analyzing the potential factors causing losses, it is possible to predict the extent to which the disaster will impact the society in the future (Vincent 2004). In order to reduce disaster losses and to improve disaster prevention capabilities, from the 1960s onward, vulnerability has formed an important research topic such as in the International Biological Program (IBP), the International Geosphere-Biosphere Programme (IGBP), the International Human Dimensions Programme on Global Environmental Change (IHDP), and the Intergovernmental Panel on Climate Change (IPCC) and so on (Zhang et al. 2008).

Socio-economic inequalities among the inhabitants are represented as a "mosaic" in space as a result of urban transformation. In addition to social-spatial isolation, such "mosaic" also leads to a redistribution of risk. Many studies on extreme events show that consequences of a disaster do not only depend on a hazard risk itself, but also are closely related to physical environments, social structures, and demographic characteristics of a geographic place (Perrow 2007; Bolin 2007). If one place is physically exposed to hazard risk, it will impact the population who live here in uneven ways (Huang et al. 2020). In urban areas, the emergence of social vulnerability is mainly determined by the instability of the internal structure of local society. Especially in the context of rapid urbanization, the continuous increase in population mobility poses severe challenges to local infrastructure, environments, and social structures. Although urban population mobility itself does not lead to vulnerability (Donner and Rodriguez 2008), the population are marginalized when the market and/or the government cannot provide adequate employment, water and sanitation facilities, housing, and medical services.

As the result of population dynamics and diverse demands for location, leading to the gradual decrease in the availability of safer lands, it is almost inevitable for human endeavors to be located in potentially dangerous places (Lavell 2003). For example, many migrants in Jakarta, Indonesia live in informal settlements called "Kampung" that are prone to flooding (Alzamil 2018). Ghana's capital Accra has 92 percent of migrants living in Old Fadama, a slum without tap-water or sanitation facilities (Awumbila 2014). The push to commercialize urban housing in China throughout the past 40 years of urbanization has widened disparities in living conditions. While existing old communities with poor living environments have not much improved, the living quality of new gated communities has significantly increased. This process has also created many marginal places, a combination of rural and urban systems characterized by high building density, unclear management rights and duties, and insufficient social infrastructure. The people who live there take the brunt of many urban disaster. Spatial and


social differentiations in the city, resulting in the formation of new socially vulnerable groups based on
various kinds of local community.
**1.2 Indicator-based Researches on Social Vulnerability**
Social vulnerability is an important indicator for evaluating such uneven regional development. It can
be understood as the ability to withstand adverse effects, the possibility of damage, and the degree of
loss caused by disasters (Timmerman 1981; Tunner et al. 2003; Cutter 1996). Meanwhile, a disaster is
not brought about only by a hazardous event but its combination with social vulnerability, and this
argument is also widely accepted by disaster researchers (Alexander 2006; Cannon 2008). Although a
single definition of social vulnerability has not been universally approved by academics, in moving
forward with researches, vulnerability has gradually developed into a widely accepted concept,
including several dimensions such as exposure, sensitivity, and adaptive capacity (IPCC 2007; IPCC
2014; Adger 2006), or exposure, resistance, and resilience (Pelling 2003).
Currently, increasing attention is paid to vulnerability in the context of climate change and urbanization.
In quantitative terms, an important goal of vulnerability assessment is to create an index of overall
vulnerability from a suite of indicators (Rygel et al. 2006). According to Parris and Kates (2003), there
have been hundreds of attempts to develop such indicators. Among them, the research that has an
important guiding role is provided by Cutter et al. (2003) focusing on Georgetown County, South
Carolina. They used county-level socio-economic and geographic statistics, divided the Social
Vulnerability Index (SoVI) into multiple dimensions, such as gender, race, age, occupation, family
structure, educational level, and so on, and revealed the vulnerability of people living in risk areas. In
the following year, with the weighted average of five sub-indices, Vincent (2004) created an index
assessing the relative vulnerability of social systems to climate change-induced variations on a
cross-national scale.
With social and environmental changes in cities, there have been increasing developments in the
quantitative assessment of vulnerability more recently. Based on the diverse ecological environments
and sociopolitical structures, many researchers (such as Rygel et al. 2006; Flanagan et al. 2011; Zhang
and You 2014; Teng et al. 2018; Xu et al. 2019) evaluated social vulnerability from different
perspectives, areas, and scopes. In doing so, they figured out relationships between vulnerability and
disasters, and tested potential risk by exploring the impact of hazards on local populations. The recent
two decades have developed other vulnerability indicators such as Environmental Vulnerability Index
(EVI) (Sopac 2004), Coastal Vulnerability Index (CVI) (Hegde and Reju 2007), Oil Vulnerability
Index (OVI) (Gupta 2008), Flood Vulnerability Index (FVI) (Balica 2007; Balica et al. 2012), and so
on. Different from many previous studies focusing on disaster losses, these studies try to examine
social vulnerability before disaster in order to identify causes of loss. By constructing indicators to
quantify vulnerability, the efficiency of communication with non-expert decision-makers has been
improved. Their main findings are compatible with disaster reduction measures, which provide a more
solid foundation for policy recommendations for disaster mitigation and preparedness.
However, most of the current social vulnerability assessments are derived from official statistics
usually in the spatial units of administrative territory. Although such macroscopic indicators of
vulnerability are of significance for disaster risk reduction at the regional level, nevertheless,
macro-data usually tend to consider general conditions, hardly reflecting locally specific conditions of
communities or individuals (You and Zhang 2013). Barnett et al. (2008) argued that indexes of


vulnerability cannot be meaningful when applied to large-scale systems, and that they should focus on
smaller scales. Especially at present, Chinese society is still under the control of the household
registration (*hukou*) system, and the large-scale floating population cannot be contained in the
macro-data. Even if the existing macro-level findings are fruitful (Teng et al. 2018), future researches
should pay more attention to the micro-level indicators of urban vulnerability, breaking through the
traditional scope, in order to have more comprehensive and in-depth results (Mao et al. 2017). Based
on previous researches, therefore, this paper selects indicators from the micro perspective, in order to
identify characteristics of urban social vulnerability and to evaluate specific groups of social
vulnerability.
**1.3 Purpose**
As China is undergoing rapid urbanization, land expansion has created different types of community
within and around the cities; the population, economy, and society are experiencing structural changes,
making the society unstable. It is imperative to mitigate the impact of disasters on urban populations
and communities, and case studies are expected to provide the policy bases for disaster risk reduction.
The main purpose of this paper is to determine the degree of social vulnerability at the local level, and
to identify the most vulnerable groups by focusing on the characteristics of social vulnerability within
Chinese urban society from the micro perspective.
This paper mainly attempts to solve the following three questions:
・What differences are in the vulnerability collectively for different types of urban communities?
・What kinds of mosaic is seen in the urban areas? That is, how vulnerable populations are distributed
across communities, and what are underlying reasons for this distribution?
・Who are the most vulnerable groups in the city, and what characteristics do they have?
This paper is organized as follows. Part Two outlines the study area from the points of geographic
location, urban development, and historical disasters. It is followed by the methodology that constructs
social vulnerability indicators weighted by expert scoring method and Analytic Hierarchy Process
(AHP). *K*-means cluster analysis is used to analyze the social vulnerability of target communities. The
results and discussions on the comparison of different communities and the identification of vulnerable
groups are then presented. Some findings are not exactly consistent with previous researches showing
that social vulnerability is rooted in specific social structural factors. This paper concludes with
suggestions for reducing social vulnerability and tackling inequality in urban China as a result of the
urbanization process.
**2 Study area**
Wuhan is a city in central China that serves as an important economic, scientific, and educational
center as well as a national transportation hub for canals, trains, highways, and flights (Figure1).
Originally, it was separated into three towns: Wuchang, Hankou, and Hanyang. After 1949, the three
towns were united into Wuhan City, which became the capital of Hubei Province in 1954. Later,




accommodating the city's growing development and population inflow, Wuhan was expanded into the
surrounding rural areas, and then divided into 13 districts (Figure 2).
Wuhan's urban population has risen steadily over the last 40 years, with the urbanization rate growing
from 47.4 percent in 1978 to 80.04 percent in 2017. The potential for population absorption continues
to rise. The city's permanent population has steadily increased in recent years, from 9.8 million in 2010
to 12.3 million in 2020, an average yearly increase of 250 thousand (Wuhan Municipal Bureau of
Statistics 2018).

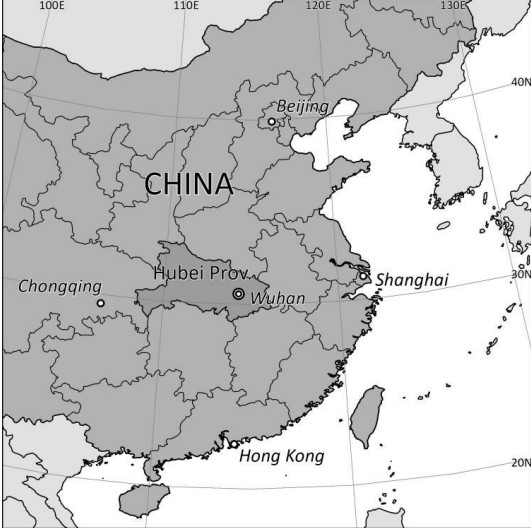

**Figure 1: The geographical location of Wuhan**
In addition, Wuhan is also one of the most vulnerable cities to natural disasters. High temperatures and
drought, heavy rains, waterlogging, freezing damage from cold temperatures, and strong winds are
some of the most common natural catastrophes. Wuhan is especially prone to extreme rains and
flooding because it has a complex internal river network, a low and flat core region, and a subtropical
monsoon climate with lots of rain.

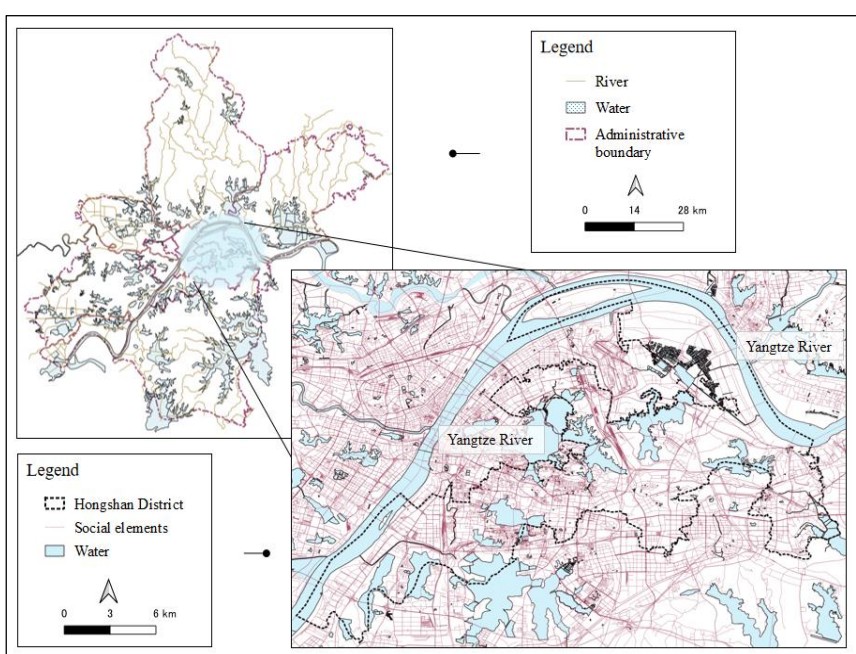

**Figure 2: Geographical features and administrative boundaries of Wuhan and Hongshan District**

Hongshan District, the target area of this study, is one of the six districts that make up Wuhan's major metropolitan area. The longest river of China, the Yangtze River, passes through Hongshan District to the southwest, flowing 75 kilometers across the district, with a water level of 14.57-20.05 meters most of the time. Floods caused by the Yangtze River's burst were common hazards to people's lives and properties prior to the year 2000. Hongshan District had 114 severe rainstorms between 1951 and 1980. The floods in 1931, 1949, 1954, 1983, 1998, and 1999 were among the most severe ever recorded (Records of Hongshan Distrist 2009). On July 21, 1998, for example, this region was hit by unprecedented and severe rains. The catastrophic flooding breach in the Hongshan district interrupted the production, and caused home collapses. There were 526 households and 103,800 people affected, and a direct economic loss of 182 million yuan for the district (Records of Hongshan Distrist 2009).

In addition to the Yangtze River, Hongshan District is surrounded by several lakes (Figure 2), with 14 lakes covering 113 square kilometers and accounting for 22.2 percent of the district's total area. In each year, the number of rainy days grows gradually from March to August. The lake level increases quickly when the rainy season begins in May, culminating in July and August. Changes in lake water levels have had a less relationship with the Yangtze River since 2000 when the dam was completed, but the main effects are from the precipitation, and the industrial, agricultural, and household water uses. As a result, the flood induced by the rising water level of the inner lakes has been the primary hazard risk in Hongshan District.

The targeted communities were chosen to represent geographically and socially local distinctions. In terms of geographic location, all the target communities are close to lakes and rivers, exposed to potential flood risks. Within China's metropolitan regions, furthermore, the housing reform policy has brought about a spatial division of labor in terms of the community's socio-economic status. According


to the explanations for the district housing plan of Wuhan city, we divided the target communities into four categories (Table 1): the community with high-grade residences (Type I), the newly demolished and rebuilt community (Type II), the old demolished and reconstructed community (Type III), and the urban villages (Type IV). Additionally, due to urbanization and land expansion, many communities are at different stages of development, which results in spatial differentiation in scenery, public facilities, and administrative management levels.

**Table 1** The types of communities

| Type | Communities | Number of respondents | Descriptions |
|---|---|---|---|
| I | G, K | 86 | Communities with high-grade residences, well-developed infrastructure, pleasant living environment, and high housing prices and rentals |
| II | A, H | 108 | Newly demolished and rebuilt communities, with the overall reasonable community planning, and higher housing prices and rents |
| III | B, C, J, I | 235 | Old demolished and reconstructed communities, with, for the most part, low-rise buildings, inadequate infrastructure, lower house prices and rents, and higher population mobility |
| IV | D, E, F | 170 | Urban villages, with poor environmental facilities, cheap rent, and a large number of migrants |

*Sources*: Records of Wuhan 1980-2000; Records of Hongshan Distrist 2009.

**3 Methodology**

For the quantitative analysis of vulnerability, identifying indicators is the first step. For many previous researches, as mentioned above, it is usual to select indicators based on external criteria such as regional economic level, infrastructure supplies level, and so on. However, there is a certain limitation that it is quite difficult if not impossible for such external criteria to grasp all aspects of the individual characteristics in any given groups. Therefore, this study rather focuses primarily on the individual ability and/or capacity that can withstand and recover from disaster in order to create a more accurate analysis of the whole spectrum of characteristics of the community.

After identifying the indicators, the next step is to weight the indicators while to analyze the vulnerable population using the data acquired from the questionnaire survey with sampled households, calculating the proportion of the high, medium, and low vulnerable populations in each type of communities. The vulnerable population often interacts with dangerousness of their living place. Thus, finally, we discuss the relationships between the vulnerabilities at the community level that are induced through the calculated 3-group proportions in each of the community types, and their social characteristics that are



provided by the explanations of the community typology, in order to get the distribution characteristics
of the vulnerable population, and to examine the new urban mosaic in Wuhan (see Figure 3).

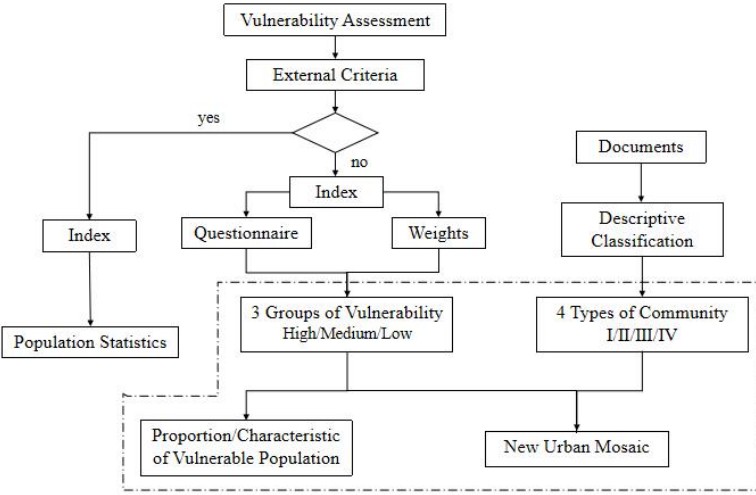

**Figure 3: The framework for vulnerability assessment**
**3.1 Selection and description of indicators**
This study selects indicators based on the concept of vulnerability, partly following the historical
disaster cases and the specific conditions of China's urban development. It adopts the IPCC's
"exposure - sensitivity - adaptive capacity" conceptual framework (IPCC 2007) as exemplified by
Füssel and Klein (2006), Füssel (2007), O'Brien et al. (2008), Coulibaly et al. (2015), Weis et al.
(2016), Fischer and Frazier (2018), to construct an evaluation index system (Table 2) and to design the
questionnaire. Although the recent vulnerability assessments following IPCC 2014 framework have
adopted the new paradigm of vulnerability that excludes exposure, this paper argues that some factors
of exposure are related to the internal state of the social system.
According to previous studies, social vulnerability is regarded as a status that exists in a certain area
prior to the disaster (Adger 2006; Bolin 2007). This status is closely related to lack of resources,
poverty, and marginalization (Hewitt 1983), and also to the adaptability of human beings to cope with
immediate or anticipated disaster pressures (Cutter 2003). As such, vulnerability index parameters are
varied depending on the objects and regions of evaluation.
*Exposure* is primarily determined by the physical location as well as the characteristics of the
surrounding built and natural environments (Pelling 2003; Perrow 2007). This research discards some
factors when choosing exposure indicators, such as the frequency of natural disasters and disaster
losses, and instead, concentrates on the location of houses, buildings, and infrastructure. Because the
locations and the built environments are interconnected to the social attributes such as social class,
income, and so forth.
Previous studies have shown that the poor may be driven to reside in hazardous regions owing to a lack





of options for location and construction since such places are less expensive (McEntire 2011). For
example, tens of thousands of largely low-income African Americans who had lived near Lake
Pontchartrain were forced to fend for themselves when Hurricane Katrina attacked the Gulf Coast of
the United States in 2005 and flooded the city of New Orleans due to breached levees (Bolin 2007).
The experts indicated that strengthening dike and flood control systems could have lessened economic
losses and saved many lives, as mentioned later. It can be seen that living in unsafe geographical
locations and buildings, and lacks of complete public facility will increase the potential exposure.
*Sensitivity* is the degree to which a system or species is affected, either adversely or beneficially by
climate variability or change, according to IPCC 2014. In a nutshell, sensitivity refers to the degree to
which the evaluated item or human is sensitive to risk, and indicates the likelihood of harm. It is
dependent on the inherent characteristics of targets (Huang et al. 2014), particularly related to
livelihood and health (Pelling 2003). Hence, to illustrate the sensitivity of the urban population, we
primarily employ population structure and economic characteristics. Previous case studies (Adger 1999;
Xu and Takahashi 2021) also showed that unstable livelihood and poor health is more sensitive to
external disturbances or changes.
*Adaptive capacity* is the ability of systems, institutions, humans to anticipate or reduce risk, to adjust to
potential damage, to take advantage of opportunities, or to respond to consequences (McCarthy et al.
2001). It is the result of the amount of intentional preparation done in light of prospective danger, as
well as spontaneous or premeditated adjustments performed in response to perceived threat (Pelling
2003). It also represents the social system through continuous adjustment of coping strategies and
measures to adapt to the surrounding environment (Klein et al. 2003). It is often influenced by
education attainment, social capital and social network (Hahn et al, 2009; Huang et al. 2014; Aldrich
2019). Individuals or groups with poorer adaptability are more likely to suffer damage and difficult to
recover from it.
In the current Chinese urban society, due to the influx of large numbers of migrants, social integration,
including social identity and self-identification, has become a key indication of rights, opportunities,
and participation. It determines individual opportunities access to resources and information. At the
same time, disaster awareness and education are required to build disaster resilience, as evidenced by
past disasters.
**Table 2**   The Evaluation Index of Social Vulnerability

| Index | Indicator | Description | Source | Positive correlation (+) or negative correlation (-) to vulnerability |
|---|---|---|---|---|
| Exposure | Geographical location | Proximity to dangerous areas such as steep slope, riverbank, sea-shore, etc. | Pelling 2003, Moss et al. 2001. | Geographical location (+) |
| | Building | Flimsy constructions unable to withstand hazard impacts. | Wisner et al. 2004 | Building fragility (+) |



| | Public infrastructure | Unavailability of critical public infrastructure. | Moss et al. 2001, Cutter et al. 2003, Vincent 2004 | Access to public facilities (-) |
|---|---|---|---|---|
| Sensitivity | Health/physical ability | Physical ability of an individual or a group of people to withstand hazard impacts. | McCarthy et al. 2001, Pelling 2003, Moss et al. 2001, Hahn et al. 2009 | Physical health (+/-) |
| | Livelihood stability | Unstable livelihoods not conducive to increasing income, easily leading to poverty. | Marshall et al. 2007 | Unstable livelihood (+) |
| | Debt | Ways of life beyond mere subsistence level and lacks of long-term investment in disaster reduction. | Ramprasad 2019 | Debt (+) |
| | Renters | Lacks of access to costly housings and of sufficient shelter options. | Cutter et al. 2003 | Renters (+) |
| | Social inclusion | No participation in local decision-making leading to social marginalization concerning social identity, self-identification, rights, opportunities, participation, etc. | Yang 2015 | Social inclusion (-) |
| | Education | Ability to understand warning information and access to recovery information. | Cutter et al. 2003, Coulibaly et al. 2015 | Low education (+) High education (-) |
| | Family structure | A large number of people under the age of 18 and over 65 depending on more energy and resources to adapt to disasters. | Vincent 2004 Hahn et al. 2009, Coulibaly et al. 2015 | Family structure (+/-) |
| Adaptive capacity | Social capital | Access to information and resources, building trust and cohesion to reduce vulnerability. | Mpanje et al. 2018, Hahn et al. 2009 | Social capital (-) |
| | Social insurance | Normal hedge against losses caused by risks, lacking the ability to overcome adverse effects. | Burton et al. 1993, McCarthy et al. 2001, IPCC 2014 | Social security (-) |
| | Social security | Sufficient social welfare to improve living conditions, thereby enhancing disaster resilience, for example pensions or allowance increasing future expectations for the younger and guarantee subsistence of the elderly. | Vincent 2004, Wisner et al. 2004, Adger and Vincent 2005 | Social welfare (-) |
| | Disaster awareness | Lack of disaster awareness and | Wisner et al. | Awareness of |



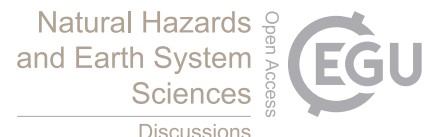

| | | experience which may impair the basic skills needed to protect oneself. | 2004 | disaster (-) |
|---|---|---|---|---|
| Disaster preparedness | | Inadequate disaster preparedness, for example food, water, rope etc., to reduce the ability to respond to disasters. | Wisner et al. 2004 | Disaster preparedness (-) |

**3.2 Determination of weight**

Weight is the relative importance of each indicator in the overall evaluation. Currently, the methodologies of determining weight can be roughly divided into subjective methods including expert scoring method, analytic hierarchy process (AHP), fuzzy comprehensive evaluation (FCE), and objective methods including entropy method, principal component analysis (PCA), factor analysis. Given the uncertainty about system dynamics (Villa and McLeod 2002; Vincent 2004), vulnerability indexes cannot be genuinely tested because they aim to provide information about the risk of future events. To be credible, the vulnerability index must either match what people actually observe in some way or at least have some intuitive resonance with experts (Sagar and Najam 1998). Therefore, this study adopts a combination of the expert scoring method and AHP to determine the weight of each indicator.

To be specific, we firstly invited ten experts from three countries, including local people with disaster experience, local scholars with disaster experience, and/or researchers on related issues in sociology and  geographers, to score 15 variables related to social vulnerability according to the degree of importance. Then, we compute the weight using the AHP with the following steps:

(1) Use the judgment matrix to calculate the weight of each indicator (including the first-level index and the second-level index), and check the consistency of the judgment matrix.

In the consistency test[1], the random consistency ratio in the judgment matrix is: $CR = \dfrac{CI}{RI}$

And the results of CR in all the matrices are less than 0.10.
(2) Calculate the final weight of each indicator.
(3) To get a more scientific result, we take the Arithmetic average, Geometric average, and Eigenvalue to calculate the weights, and then regard the average as the final weight of each indicator (Table 3).

**Table 3** The weight of Indicators

| Index | Weight | Indicator | Weight | Final weight |
|---|---|---|---|---|
| Exposure | 0.5394 | Geographical location | 0.3334 | 0.179836 |
| | | Building | 0.5689 | 0.306865 |

---

[1] Consistency ratio (CR); Consistency index (CI); Random consistency index (RI)





| | | Critical infrastructure | 0.0977 | 0.052699 |
|---|---|---|---|---|
| Sensitivity | 0.1635 | Health/physical ability | 0.491 | 0.080279 |
| | | Livelihood stability | 0.3056 | 0.049966 |
| | | Debt | 0.1254 | 0.020503 |
| | | Renters | 0.078 | 0.012753 |
| Adaptive capacity | 0.2971 | Social inclusion | 0.0454 | 0.013488 |
| | | Education | 0.0454 | 0.013488 |
| | | Family structure | 0.0454 | 0.013488 |
| | | Social capital | 0.1887 | 0.056063 |
| | | Social insurance | 0.075 | 0.022283 |
| | | Social security | 0.1189 | 0.035325 |
| | | Disaster awareness | 0.2925 | 0.086902 |
| | | Disaster preparedness | 0.1887 | 0.056063 |

**3.3 Data collection and analysis**
The preliminary interviews and the questionnaire surveys were conducted in June and July of 2021,
respectively. First, we designed the questionnaires using the social vulnerability index and the
preliminary interviews with local residents. In addition, when selecting the sampling method, it was
taken into account that many urban migrants, especially low-skilled and low-secured representatives of
migrant workers, were not fully included in the urban population list. Therefore, we adopted the
method of quota sampling to determine the sample size of each community, and the questionnaires of
each community were obtained by random survey. A total of 620 questionnaires (including 599 valid
responses, an effective rate of 96.6%) were collected from 11 communities (from A to K) in 8 streets of
Hongshan District, Wuhan City (see Table 1).
In order to eliminate the influence of different dimensions and orders of magnitude, we adopt
normalization to deal with each index. Min-max normalization is used to make the numerical value of
all indexes between 0 and 1.
Normalization for positive indicators:

$$x'_{ij} = \frac{x_{ij} - min\ \{x_j\}}{max\ \{x_j\} - min\ \{x_j\}}$$

Normalization for negative indicators:

$$x'_{ij} = \frac{max\ \{x_j\} - x_{ij}}{max\ \{x_j\} - min\ \{x_j\}}$$

$x_{ij}$ represents the value of the $j$th index of the $i$th surveyed object and $min\ \{x_j\}$ and $max\ \{x_j\}$
represent the minimum and maximum value of the $j$th index of all the surveyed objects respectively.



The vulnerability value can be calculated after the normalization process.
6                                    **Table 4**    The normalized variables

| Serial number | Variable | Maximum | Minimum | Mean value | SD |
|---|---|---|---|---|---|
| 1 | Geographical location | 1 | 0 | 0.4372 | 0.1982 |
| 2 | Building | 1 | 0 | 0.4265 | 0.2103 |
| 3 | Critical infrastructure | 1 | 0 | 0.5245 | 0.2063 |
| 4 | Health/ Physical ability | 1 | 0 | 0.2872 | 0.2594 |
| 5 | Livelihood stability | 1 | 0 | 0.3863 | 0.2852 |
| 6 | Debt | 1 | 0 | 0.1957 | 0.5076 |
| 7 | Renters | 1 | 0 | 0.4599 | 0.5402 |
| 8 | Social inclusion | 1 | 0 | 0.2772 | 0.1788 |
| 9 | Education | 1 | 0 | 0.6064 | 0.2819 |
| 10 | Family structure | 1 | 0 | 0.3871 | 0.2877 |
| 11 | Social capital | 1 | 0 | 0.4526 | 0.2078 |
| 12 | Social insurance | 1 | 0 | 0.6614 | 0.3023 |
| 13 | Social security | 1 | 0 | 0.4603 | 0.2578 |
| 14 | Disaster awareness | 1 | 0 | 0.5004 | 0.1647 |
| 15 | Disaster preparedness | 1 | 0 | 0.7051 | 0.2973 |

In order to compare the social vulnerability of target communities and identify the characteristics of
vulnerable groups, in this paper, *K*-means cluster analysis was adopted to divide the vulnerability
values into three categories of high, medium, and low. Cluster analysis is a statistical method that
divides research objects into reasonably homogeneous groups. The same cluster of levels of social
vulnerability is a reflection of the similar ability of individuals and communities to withstand risks, and
its level directly means the possibility of individuals or communities succumbing to disasters.
Quantitative (discrete and continuous) variables reveal the current vulnerability of the Wuhan
communities as well as the probability that they may be affected by disasters in the future.
**4 Results and Discussion**
**4.1 Comparison of Different Communities' Social Vulnerability**
Eleven communities from A to K are divided into four categories of Types I to IV based on their states
of development, in terms of their built environments, demographic compositions, housing prices, and
other features (Table 1). The social vulnerability of these four types of communities is each calculated,




and it is shown that there are significant disparities in vulnerability between them (Figure 4).
Type I communities have the lowest social vulnerability, followed by Types II and III, while that of
Type IV communities with the highest value. Moreover, the four types of communities have
statistically significant differences in their levels of vulnerability (see Figure 4).

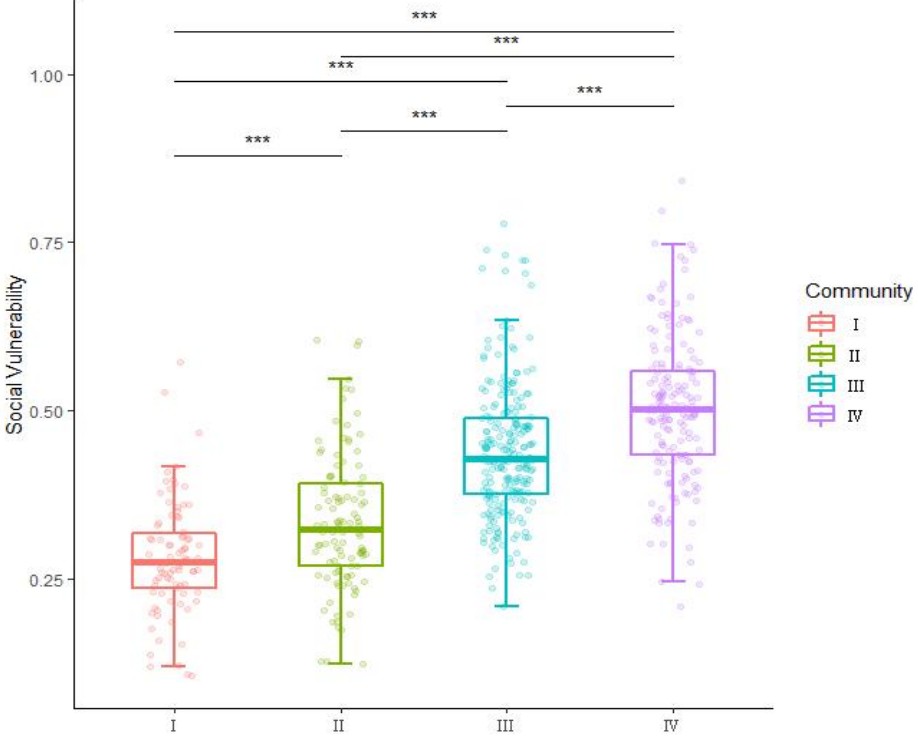

**Figure 4: Social Vulnerability Box Plot of 4 type communities**   *Note*: p < .01*** (= .000)
Figure 4 also shows that Type I communities have the most concentrated distribution of vulnerability,
implying that the vulnerability gap among individuals in each community of Type I is the smallest.
According to the survey data, their residents are homogeneous in socioeconomic traits such as
educational attainment and income stability.
The most dispersal data of Type IV communities indicates that the disparity of individuals'
vulnerabilities in Type IV communities is relatively large, and the fact is related to the high rate of
floating population in urban villages as well as the heterogeneity of population attributes and social
characteristics. Types Ⅱ and Ⅲ communities are rebuilt after demolition and relocation, referred to as
*Huanjianfang* in Chinese. The so-called *Huanjianfang* refers to the government demolition of the
original houses of farmers in the suburban areas for the purpose of municipal construction, then
accommodating the new houses. New dwellings are reallocated to residents who have demolished their
original houses in the form of compensation. It is in a unique process of dwelling in China's
urbanization process, and subject to some restrictions related to the circulation. Furthermore, in order to
save costs, developers frequently utilize inferior building materials. The main difference between two

is that the communities of Type II are superior to those of Type III in terms of housing density, construction quality, infrastructure and greening. As a result, despite the fact that both types are rebuilt following the renewal of former villages in the rural-urban fringes, there is still a significant disparity in the characteristics and vulnerabilities of the people between the two types.

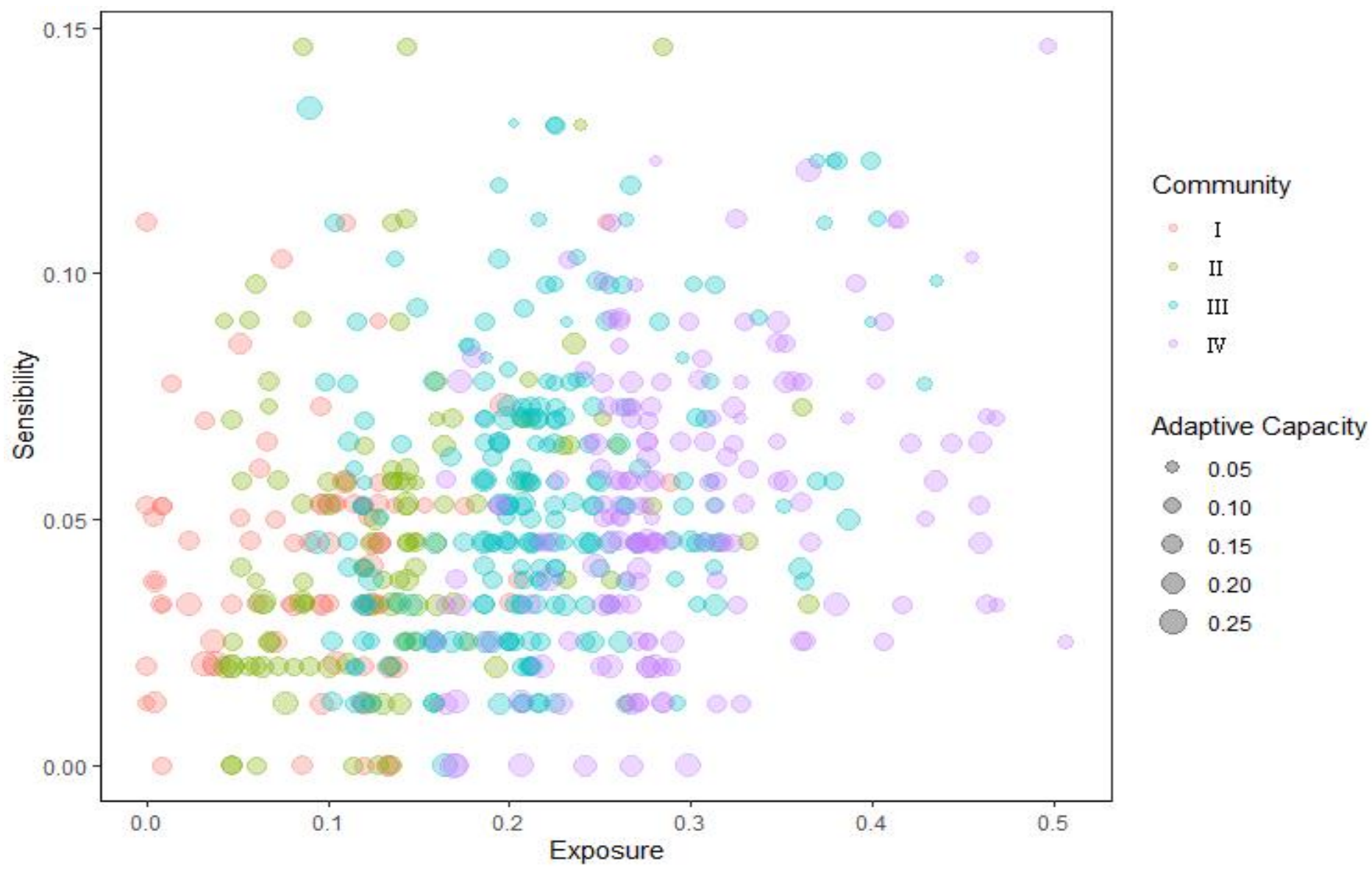

**Figure 5: Bubble chart of exposure, sensitivity, and adaptive capacity**

In comparison to sensitivity and adaptability as dimensions of vulnerability, exposure fluctuates the most. Types I and II of communities are significantly less exposed than Types III and IV, with the fourth type seeing the most exposure, namely in dangerous geographical and physical conditions. The difference in sensitivity across four types is minor, with most of the people in Types I and II being somewhat less sensitive than those in Types III and IV; but individuals within each group, on the other hand, differ significantly. A previous study (Turner et al. 2003) found that not only do social vulnerabilities vary between societies, communities, and groups, but also among residents in the same area/community. We have verified that using quantitative analysis receives similar findings (see Figure 5).

Although the majority of highly exposed and highly sensitive individuals are also showing poor adaptive capacity, the four types of communities have very little variation in individual adaptability, and the aggregate values are not all that high, according to the bubble chart. Furthermore, Figure 5 demonstrates that overall sensitivity and adaptability have a negative relationship. More sensitive people are less adaptive. Adaptability, on the other hand, improves when sensitivity decreases.

*4.2 Social vulnerability and residential segregation*



As the result of the cluster analysis gaining three categories of high, medium, and low groups for the
individual vulnerabilities, the group of high vulnerability accounts for 12.9 percent of the 599 samples
investigated, the group of medium vulnerability for 48.4 percent, and the group of low vulnerability for
38.7 percent, respectively. Eventually, the social vulnerability in the study area is moderate for almost
the half, with a much lower proportion of high vulnerability.
**Table 5** The distribution of individuals social vulnerability

| Level of vulnerability | Percentage of individuals in 4 type communities | | | | | Numerical range |
|---|---|---|---|---|---|---|
| | I | II | III | IV | Total | |
| High-vulnerability | 1 (11) | 3 (14) | 26 (30) | 47 (22) | 77 | [0.5488, 0.8416] |
| | 1.3% | 3.9% | 33.8% | 61.0% | 100% | |
| Medium-vulnerability | 10 (42) | 28 (52) | 150 (114) | 102 (82) | 290 | [0.3772, 0.5478] |
| | 3.4% | 9.7% | 51.7% | 35.2% | 100% | |
| Low-vulnerability | 75 (33) | 77 (42) | 59 (91) | 21 (66) | 232 | [0.1055, 0.3767] |
| | 32.3% | 33.2% | 25.4% | 9.1% | 100% | |
| Total | 86 | 108 | 235 | 170 | 599 | |
| | 14.4% | 18.0% | 39.2% | 28.4% | 100% | |

$X^2$ (6, N =599) =222, p < .01*** (= .000); the figures in ( ) are expected values.

From Table 5, it can be found that there are a few individuals classified into high-vulnerability and
medium-vulnerability groups in the communities of Types I and II. More than 90 percent of the highly
vulnerable groups and more than 85 percent of the moderately vulnerable groups are concentrated in
the communities of Types III or IV. Almost half of the moderately vulnerable groups are in Type III;
the communities of Type IV, thought of as an urban village, are mainly composed of individuals
classified into high vulnerability group; a few individuals of low-vulnerability group.
Furthermore, when comparing the vulnerability characteristics between the community types (Figure 6),
it is not difficult to see that, while    communities of Type III have fewer scores than those of Type IV
in terms of exposure and adaptive capacity, higher in sensitivity. The communities of Type III are
thought of as transitioning from urban village to urban community. The population here is confronted
with many unpredictable circumstances, and changes in expectations for the future may have an impact
on their ability and stability, leading to an increase in sensitivity and a loss of potential for adaptation
(Figure 6). Moreover, when such a twilight district as an urban village is demolished, its communities
quickly lose their relative geographical and environmental advantages and the people are compelled to
relocate. Their low income will provide not many options for where to reside, thus being forced into
more exposed neighborhoods, with a high likelihood of becoming a high-vulnerability population.

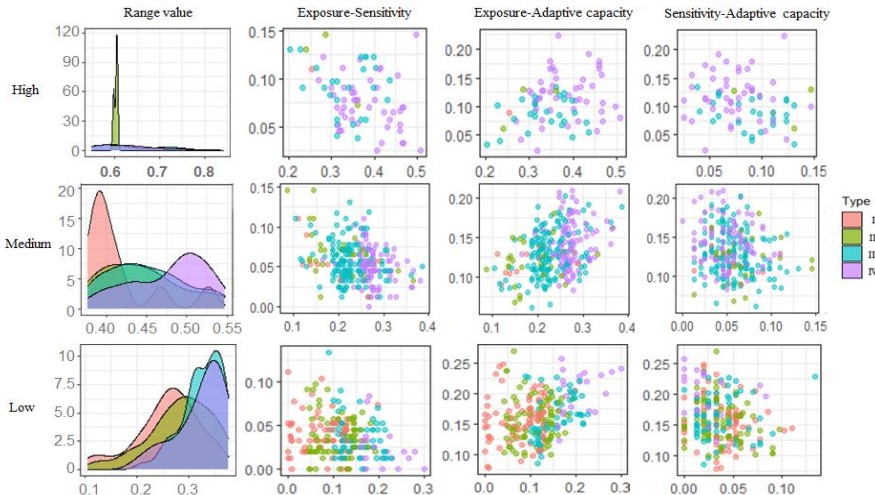

**Figure 6: The distribution and characteristics of high, medium and low-level vulnerability**
The disparity in social vulnerability among inhabitants in various neighborhoods implies "residential
segregation" in the metropolitan environments. An urban community is more than just a "geographic
location," but also a physical and social environment. Urban residents' occupations, incomes,
household registrations (*hukou*), and educational backgrounds differ accordingly, as does the
affordability and need for living space and supporting public service facilities among them.
The rapid urbanization of Chinese cities over the past four decades has generated a new socio-spatial
disparity. This socio-spatial disparity has shattered the initial social homogeneity that had existed
before the reform and opening up of the 1980s. There is a growing tendency to polarize urban districts
as well as to increase degrees of intra and inter-neighborhood segregation. Low-income groups and
floating populations frequently relocate within cities in order to find better jobs and more affordable
housing. Only when they can gain access to economically favorable environments with lower rent by
moving to dangerous places, they relocate to such places regardless of disaster risks (Hardoy and
Satterthwaite 1989). Households or individuals without any financial capacity to afford
minimum-standard housing are forced to make some compromises, often with preference for food for
the family and education for children (Hardoy and Satterthwaite 1987).
Even though the communities of Types I II are geographically close to lakes and rivers, these types of
communities outperform other communities in terms of the built environment which also influences the
vulnerability (Pelling 2003). On the one hand, a high-quality building environment, comprising solid
housing, appropriate provisions for waste collection and sanitary disposal, a full fire protection system,
and so on results in expensive housing prices, which excludes the majority of low-income groups. The
increase in rents caused by the successive demolition and reconstruction of twilight urban districts in
the municipal planning has forced them to find affordable housing elsewhere. This is the reason why
high-vulnerability and medium-vulnerability residents are concentrated in the communities of Types III
and/or IV. On the other hand, the unfavorable conditions in housing, medical care, job opportunities,
and public services, which may hinder or limit the residents' access to high-quality resources and


opportunities, exacerbate their precarious situations, and weaken their abilities to withstand disasters.
This is why the overall social vulnerability of residents in the third and fourth types is higher than that
of other type communities.
In this sense, such social segregation is projected onto space (Cassiers and Kesteloot 2012) and implies
an overlap of dual marginalization in the spatial and social terms. Social vulnerability develops through
a process of socio-spatial and intraurban heterogeneity. Many factors, such as poverty, and poor
housing and infrastructure, lead to disparities in the social vulnerability of diverse communities and
groups. They may suffer differentiation of shocks and losses, in the case of a future calamity.
*4.3 Identification of vulnerable populations*
The difference in the social vulnerability of different communities is an indirect reflection of
socio-spatial divergence and a manifestation of the polarization between urban affluent and poor
groups. Differentiated groups' social vulnerabilities are caused by structural factors in the society,
which are derived from the features of the system (Clark et al. 2000). Residents in cities have different
groups due to their different economic statuses, cultural backgrounds, living conditions, and other
comprehensive factors. The relevant factors of social vulnerability are helpful in the identification of
vulnerable groups and the implementation of particular attention and protective strategies for them.
**Table 6**   Social characteristics of individuals with different vulnerabilities

| | Trait | Description | Mean value | Low | Medium | High |
|---|---|---|---|---|---|---|
| | Age | - | 45.2037 | 43.4353 | 46.5828 | 45.3377 |
| Personal factors | Education | 1 Elementary school and below<br>2 Junior high school<br>3 Senior high school<br>4 Junior college<br>5 Undergraduate<br>6 Postgraduate and above | 2.9666 | 3.3276 | 2.7586 | 2.6623 |
| | Health | 1 Very poor<br>2 Poor<br>3 General<br>4 Well<br>5 Very well | 3.8531 | 4.2500 | 3.7621 | 3.0779 |
| Economic factors | Personal annual income | 1 Under 25000<br>2 25000-50000<br>3 50000-75000<br>4 75000-100000<br>5 Over 100000 | 2.2337 | 2.4483 | 2.1276 | 1.9870 |
| | Livelihood | 1 Very low stable | 3.4558 | 3.8060 | 3.3586 | 2.7662 |






| | stability | 2 Low stable | | | | |
| | | 3 Stable | | | | |
| | | 4 High stable | | | | |
| | | 5 Very high stable | | | | |
| Social factors | Social inclusion | 1 Be excluded completely | 3.8932 | 4.0862 | 3.8483 | 3.4675 |
| | | 2 Be excluded | | | | |
| | | 3 General | | | | |
| | | 4 Be involved | | | | |
| | | 5 Be fully involved | | | | |
| | Social Security | 1 None | 3.1569 | 3.4871 | 3.0207 | 2.6364 |
| | | 2 Insufficient | | | | |
| | | 3 General | | | | |
| | | 4 Sufficient | | | | |
| | | 5 High sufficient | | | | |
| | Social insurance | 1 None | 2.3539 | 2.9224 | 2.0724 | 1.8182 |
| | | 2 Insufficient | | | | |
| | | 3 General | | | | |
| | | 4 Sufficient | | | | |
| | | 5 High sufficient | | | | |

Judging from the mean value of the characteristics in Table 6, individuals with high-vulnerability have traits such as a low level of education and health, a low level of annual income, and unstable work. Especially there are substantial discrepancies between high- and low-vulnerable groups of individuals in terms of health status, job stability, and social insurance.

There is a little gap between medium- and high-vulnerable groups of individuals in terms of education, annual income, and social insurance, but a large discrepancy in health status and employment stability. This indicates relative high sensitivity of the populations of medium-vulnerability. They are more prone to slip into high-vulnerability if their physical health and livelihood security is jeopardized by external pressure.

The average age of the low-vulnerability group is lower than the sample average, but it is somewhat higher than that of the medium-vulnerability group, showing their not clear interrelationships. Despite the disadvantages of the elderly in terms of their physical conditions, we can argue that they often have a relatively high level of social security, as well as other aspects such as wealth accumulation, income stability, and living conditions that are superior to most of younger people in urban China, the situations of which may be different from rural China. As a consequence, even if previous researches have pointed out that higher vulnerability is seen in older groups, the findings of this study differ from it. It is indispensable to judge based on social backdrop and development level, when developing indices of vulnerability assessment.

There are also other categorical factors, such as occupation, household registration, gender, and debt, in addition to the continuous variables listed above. Because the value of these variables cannot reflect the variations in individual social vulnerability, they must be examined independently.



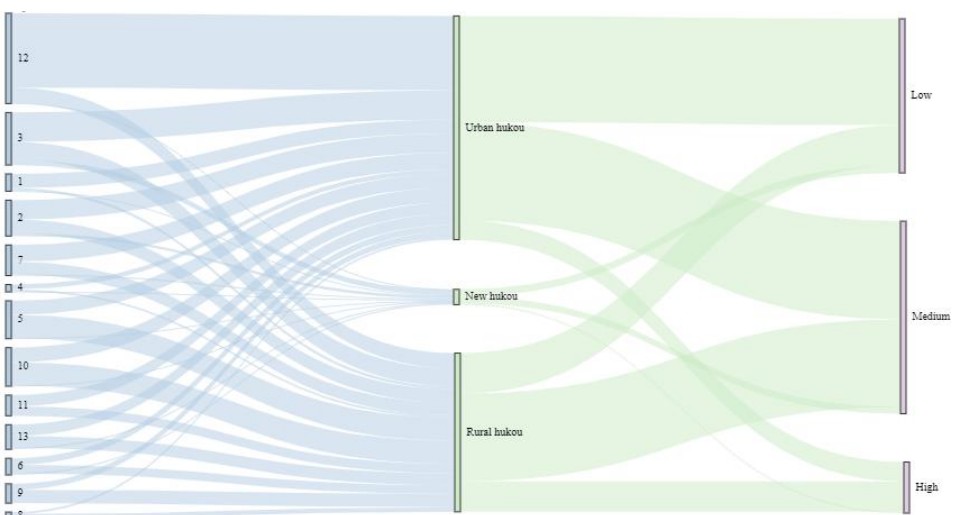

$X^2$ (24, N =599) =98.63, p < .01*** (= .000)        $X^2$ (4, N =599) =34.370, p < .01*** (= .000)
**Occupation (on the left bar)**: 1=Staff of governmental departments and institutions, 2=Professional
and technical personnel, 3=Company employees, 4=Businessmen, 5=Service personnel in the tertiary
sector, 6=Industrial workers, 7=Students, 8=Agricultural workers, 9=Housewives, 10=Private business

6                owner, 11=Unemployed, 12=Retired person, and 13=Other.

**Figure 7: Correspondence between occupation(on the left bar), household registration (hukou) (on the**
**middle bar) and social vulnerability level (on the right bar)**
From Figure 7, in terms of the type of *hukou*, the high vulnerability can be seen more in the group of
rural *hukou* holders than in that of urban *hukou*. Among the high-vulnerability groups approximately
60% hold rural *hukou*, accounting for half of the medium-vulnerability group. People primarily
employed in service industries, self-employed, and low-skilled workers make up the majority of
rural-to-urban migrants looking for better employment prospects. Low-skilled workers lack adequate
social security, and their income stability has always been in jeopardy. As for the self-employed and
those in the service industry like receptionists, waiters, call-center employees, it is likely that their
livelihoods have also fallen into instability as seen in the impacts of the recent pandemics and the
following city lockdowns in Wuhan. Most of them have low incomes, live in densely populated poor
communities or urban villages, and lack comprehensive social welfare programs. These are the main
reasons for their higher vulnerability.
Although there are also some low-vulnerability individuals with the rural household registration, it can
be argued that they are mainly engaged in state-owned enterprises, including public service units. Such
jobs have high stability in terms of income and social security. Among them, enterprises and units with
better social welfare may provide the opportunity for urban *hukou* holders (called *Luohu* in Chinese).
Moreover, higher education, stable wealth accumulation, social status, and so on can contribute to the
transformation from rural *hukou* to urban *hukou*, as the origin of urban *hukou* of a new citizen.
Following the acquisition of local urban *hukou*, they will benefit in the same way as local urban
residents.


China's household registration system, *hukou*, which is an institution controlling population movement, to a certain extent represents social and economic outcomes at the individual level (Liu 2005). Entitlements to state-supplied social benefits and opportunities including education and medical service, and social security benefits including unemployment insurance, endowment insurance, and housing security are still rationed based on household registration. Therefore, migrants without local urban *hukou* usually face difficulty to access local public services and social security benefits in a city. Megacities are particularly challenged in this regard. But the decline in *hukou*'s influence on career choices can also be seen from Figure 7. Indeed, not a few rural-to-urban migrants with rural *hukou* are no longer engaged in low-end labor and temporary jobs as they came approximately 20 years ago (see Chan and Zhang 1999), and now they have more choices in careers. However, there remains a problem that they are still unable to enter high-paying and stable industries, and the impact of *hukou* on individual social vulnerability cannot be ignored.

At the same time, the results also show that about 50% of urban registration holders are also at high and medium levels of social vulnerability. Many studies have so far argued that China has the problem of unequal distribution of resources between urban and rural areas at the national level, and that urban residents have advantages in the acquisition and utilization of various resources (Sicular et al. 2007; Liu et al. 2019). Relatively speaking, the inequality within the urban population has not received much attention. In fact, a large part of the urban population, due to various reasons resulting in poverty and lack of opportunity, exhibits insufficient resilience and resistance to disasters in facing dangers, shocks, and pressures. Although social vulnerability cannot be read directly off from poverty (Chambers and Conway 1992), the former is often very highly interrelated to the latter (Wisner et al. 2004), causing such inequality.

At present, most of the urban poor in China are not absolutely but relatively poor, and the gap between the rich and the poor is constantly widening. China's Gini Coefficient[2] from 2003 to 2017 was between 0.462 and 0.491 (National Bureau of Statistics 2018), indicating increasingly inequality of income. In addition to the income gap, differences in assets are creating much more inequality. With the development of urbanization, the poor will be poorer in urban areas, and the rich will be richer. There is no opportunity for upward mobility in the lower classes of the city, and the mobility between various strata of Chinese society has been significantly reduced, implying social hierarchical consolidation. With the widening income gap, poverty may be spreading rapidly throughout the cities, as is vulnerability within the cities as well. Some systems of society have inherent forces creating inequalities (Mehretu et al. 2002), the macro data may hide these inequalities, making the scale and depth of urban vulnerability underestimated.

**5 Conclusion**

Through the development of micro-individual social vulnerability indicators and the use of cluster analysis, this research has assessed the level of social vulnerability of 599 residents in 11 communities

---

[2] It is generally believed that the income of residents is very average when the Gini coefficient is less than 0.2, It is generally believed that the income of residents is very average when the Gini coefficient is less than 0.2, average between 0.2 and 0.3, more reasonable between 0.3 and 0.4, and the gap between 0.4 and 0.5 is too large, and when the gap is greater than 0.5, the gap is huge.





in Hongshan District of Wuhan. The findings show three levels of social vulnerability: high, medium,
and low. Quantitative assessments offer comparisons specifically between distinct units and, the results
indicate that different types of communities have great differences in social vulnerability. Residents of
favorable communities have more resources and opportunities, and because of this, they have an option
of living in areas with comparably superior conditions. Therefore, they have lower exposure and
sensitivity, and higher adaptability to disaster risks. But inhabitants in urban villages are in a different
scenario. Residential segregation is an important consideration from the results of assessing social
vulnerability. Another main finding is that higher vulnerability groups have the characteristics of low
education, poor health, low annual income, unstable work, and insufficient social security. Improving
the stability of livelihoods, wealth accumulation, social security, and so on contributes positively to
reducing individual social vulnerability.
The aforementioned socio-spatial differences are not confined to Wuhan or Chinese cities, but also
exist in other parts of the world, in developed cities like New York or emerging cities like Jakarta.
When inequality reaches a certain level, it will trigger social crises. Whether we live in nations with
robust or weak economies, structural inequality will reveal itself during crises, harming those who are
already impaired and defenseless (Kalpana Sharma 2020). Despite the fact that climate change and
urbanization are worldwide phenomena, impoverished people and disadvantaged groups are
disproportionately affected due to factors such as poverty, excessive reliance on natural resources, and
inadequate infrastructure. Returning to the case of China, to minimize the social vulnerability generated
by the urbanization process, underlying inequalities within the city must be addressed. First, measures
should be implemented to ensure housing and social security that might be reduced for example by
controlling housing prices and constructing public housing. Solving the *hukou* problem, which causes a
disparity in benefit between residents with and without urban *hukou*, could achieve social security
justice. Second, in order to effectively manage hazard risks and decrease disaster losses, we must take
into account different groups when developing climate adaptation and urban development policies,
particularly disadvantaged individuals at the bottom of society who have no voice.
The importance of this research in terms of practical application is twofold: first, it constructs
individual-scale indexes and analyzes vulnerability using existing indicators for different spatial scales
and groups, which contributes to the research on micro-vulnerability indicators in China's cities
lacking basic micro-level statistics. The second quantitative analysis properly assesses and
comprehends the most vulnerable groups, allowing for community comparisons. This helps policies be
undertaken to support the most vulnerable communities and the most vulnerable population.
Nonetheless, we must acknowledge that social vulnerability in the context of urbanization is a complex
issue that is the result of numerous variables interacting and impacting one another. It is also a major
development issue that affects economic and social progress, as well as human security and well-being.
More microscopic social vulnerability indicators that can represent reality might need to be explored in
future studies. It is equally important to investigate how social vulnerability is (re)produced. The most
essential humanistic care is to focus on poor neighborhoods and vulnerable populations. Passive
avoidance is not an option for both regular people and especially the government. Action must be taken
to safeguard them in order to reduce their vulnerabilities.



**Appendix A:    Detailed Calculation for correspondence between occupation, household**
**registration (*hukou*), and social vulnerability level (See Figure 7)**
**Table A1**    *Hukou* and Social Vulnerability

*Hukou* and Social Vulnerability

|  |  | High | Medium | Low | Total |
|---|---|---|---|---|---|
| *Hukou* | Urban *hukou* | 160 (131) | 148 (163) | 29 (43) | 337 |
|  | Rural *hukou* | 61 (93) | 132 (116) | 46 (31) | 239 |
|  | New *hukou* | 11 (9) | 10 (11) | 2 (3) | 23 |
| Total |  | 232 | 290 | 77 | 599 |

$X^2$ (4, N =599) =34.370, p < .01*** (= .000)

**Table A2**    Occupation and *Hukou*

Occupation and    *Hukou*

|  |  | Urban *hukou* | Rural *hukou* | New *hukou* | Total |
|---|---|---|---|---|---|
| Occupation | 1 | 21 (15) | 4 (10) | 1 (1) | 26 |
|  | 2 | 29 (30) | 21 (22) | 4 (2) | 54 |
|  | 3 | 44 (44) | 27 (32) | 8 (3) | 79 |
|  | 4 | 7 (6) | 3 (4) | 1 (0) | 11 |
|  | 5 | 21 (32) | 35 (23) | 1 (2) | 57 |
|  | 6 | 10 (14) | 13 (10) | 2 (1) | 25 |
|  | 7 | 25 (26) | 19 (18) | 2 (2) | 46 |
|  | 8 | 3 (6) | 7 (4) | 0 (0) | 10 |
|  | 9 | 9 (16) | 20 (12) | 0 (1) | 29 |
|  | 10 | 22 (33) | 35 (23) | 1 (2) | 58 |
|  | 11 | 17 (17) | 14 (12) | 0 (1) | 31 |
|  | 12 | 112 (77) | 23 (54) | 1 (5) | 136 |
|  | 13 | 17 (21) | 18 (15) | 2 (1) | 37 |





| Total | 337 | 239 | 23 | 599 |
|---|---|---|---|---|

$X^2$ (24, N =599) =98.63, p < .01*** (= .000)

Notes:

1=Staff of governmental departments and institutions    2=Professional and technical personnel    3=Company employees

4=Businessmen    5=Service personnel in the tertiary sector    6=Industrial workers    7=Students    8=Agricultural workers

9=Housewives    10=Private business owner    11=Unemployed    12=Retired person    13=Other

**Data availability:** The data and analysis code are available by contacting the corresponding author.

**Author Contributions:** JX and MT conceptualized the work. JX, MT, WFL developed the model. WFL, JX organized the questionnaire survey and conducted the quantitative analysis. The project administration and funding acquisition from MT. JX provided original draft preparation. JX and MT reviewed and edited the paper. All authors visualized the data.

**Declaration of competing interest:** The authors declare that they have no known competing interests or personal relationships that could have appeared to influence the work reported in this paper.

**Disclaimer:** Publisher's note: Copernicus Publications remains neutral with regard to jurisdictional claims in published maps and institutional affiliations.

**Acknowledgments:** The authors would like to express the gratitude to students of Huazhong Agricultural University for their participation in the questionnaire survey. We also thank for feedback from all the respondents.

**Financial support:** The research is mainly supported by JSPS Grant-in-Aid for Scientific Research (B), Project Number 19H01381, with the second author as a principle investigator. The first author acknowledges Fundamental Research Funds for the Central Universities of China under Grant DUT22RC(3)089 to provide the material support. Technical support is partly from Fundamental Research Funds for the Central Universities of China under Grant 2662020LXQD002, with the third author as a principle investigator.



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

Index for Disaster Management. Journal of Homeland Security and Emergency Management,

8, 1-22, https://doi.org/10.2202/1547-7355.1792, 2011.

Fischer A.P., Frazier T.G.: Social vulnerability to climate change in temperate forest areas: New

measures of exposure, sensitivity, and adaptive capacity, Annals of the American Association

of Geographers, 108, 658-678,

https://doi.org/10.1080/24694452.2017.1387046, 2018.

Füssel Hans-Martin, Klein Richard J.T.: Climate Change Vulnerability Assessments: An

Evolution of Conceptual Thinking, Climatic Change, 75, 301-329,


https://doi.org/10.1007/s10584-006-0329-3, 2006.
Fussel, H.: Vulnerability: A generally applicable conceptual framework for CC research, Global
Environmental Change, 17,155-167,
https://doi.org/10.1016/j.gloenvcha.2006.05.002, 2007.
Gupta, E.: Oil vulnerability index of oil-importing countries, Energy Policy, 36,1195-1211,
https://doi.org/10.1016/j.enpol.2007.11.011, 2008.
Hahn, M. B., Riederer A. M., and Foster S. O.: The livelihood vulnerability index: a pragmatic
approach to assessing risks from climate variability and change: a case study in Mozambique,
Global Environmental Change, 19, 74-88,
https://doi.org/10.1016/j.gloenvcha.2008.11.002, 2009.
Hardoy Jorge E. and Satterthwaite D.: Third world cities and the environment of poverty,
World Health Forum, 8, 86-93, https://apps.who.int/iris/handle/10665/47357, 1987.
Hardoy Jorge E. and Satterthwaite D.: Squatter Citizen: Life in the Urban Third World,
London, Earthscan, the United Kingdom, ISBN-10 1853830208 and
ISBN-13978-1853830204,1989.
Hegde V.A., and Reju R.V.: Development of Coastal Vulnerability Index for Mangalore Coast,
India. Journal of Coastal Research, 23, 1106-1111,
https://doi.org/10.2112/04-0259.1, 2007.
Klein J. T.Richard, Nicholls J.Robert, Thomalla F.: Resilience to natural hazards: How useful is
this concept? Environmental Hazards, 5, 35-45,
https://doi.org/10.1016/j.hazards.2004.02.001, 2003.
Hewitt, K.: Interpretations of Calamity from the viewpoint of human ecology, London, Allen and
Unwin, the United Kingdom, ISBN 0-04-301160-8, 1983.
Huang X.J., Huang X., and Cui C.L.: The concept, analytical framework and assessment method
of social vulnerability, Progress in Geography, 33, 1512-1525, 2014.
Huang X.J., Wang B., Liu M.M., Guo Y.H. and Li Y.Y.: Characteristics of urban extreme heat
and  assessment of social vulnerability in China, Geographical Research,39, 1534-1547,

28     2020.

IPCC, 2007: Climate Change 2007. Impacts, Adaptation and Vulnerability. Contribution of
Working Group II to the Fourth Assessment Report of the Intergovernmental Panel on
Climate Change [Parry, M.L., O.F. Canziani, J.P. Palutikof, P.J. Van Der Linde, and C.E.
Hanson (eds.)]. Cambridge University Press, Cambridge, UK, pp. 7-22. (last access: 15 May

33    2021)

IPCC, 2012: Summary for Policymakers. In: Managing the Risks of Extreme Events and Disasters
to Advance Climate Change Adaptation [Field, C.B., V. Barros, T.F. Stocker, D. Qin, D.J.
Dokken, K.L. Ebi, M.D. Mastrandrea, K.J. Mach, G.-K. Plattner, S.K. Allen, M. Tignor, and
P.M. Midgley (eds.)]. A Special Report of Working Groups I and II of the Intergovernmental
Panel on Climate Change. Cambridge University Press, Cambridge, UK, and New York, NY,
USA, pp. 1-19. (last access: 15 May 2021)
IPCC: Climate Change 2014: Synthesis Report. Contribution of Working Groups I, II and      III
to the Fifth Assessment Report. Geneva, Switzerland: Intergovernmental Panel on Climate
Change, 2014, 151. (last access: 15 May 2021), 2014.



Lavell, A.: Local Level Risk Management: Concept and Practices. CEPREDENAC-UNDP,
Quito, Ecuador, 2003.
Liu, Z.: Institution and inequality: the hukou system in China. Journal of Comparative
Economics, 33, 133-157, https://doi.org/10.1016/j.jce.2004.11.001, 2005.
Mao Y.H.,Yu D.L.,Zheng J.H., Wang H.L.: Progress and research of urban vulnerability,
Environmental Science & Technology, 40, 97-103, 2017.
Marshall, N. A., Fenton, D. M., Marshall, P. A., Sutton, S. G.: How resource dependency can
infuence social resilience within a primary resource industry, Rural Sociology, 72, 359-390,
https://doi.org/10.1526/003601107781799254, 2007.
McCarthy, J.J., O.F. Canziani, N.A. Leary, D.J. Dokken, and K.S. White:
Climate Change 2001: Impacts, Adaptation, and Vulnerability, Cambridge,Cambridge
University Press,the United Kingdom, ISBN 0-521-01500-6 and ISBN 0-521-80768-9, 2001.
McEntire, D.: Understanding and reducing vulnerability: from the approach of liabilities and
capabilities, Disaster Prevention and Management, 20, 294-313.
https://doi.org/10.1108/09653561111141736, 2011.
Mehretu, A., Pigozzi, B.Wm., Sommers, L.M.: Concepts in social and spatial marginality,
Geografiska. Annaler, Series. B Human. Geography, 82, 89-101,
https://doi.org/10.1111/j.0435-3684.2000.00076.x, 2003.
Moss, R.H., Brenkert, A.L. and Malone, E.L.: Vulnerability to Climate Change, A
Quantitative Approach. Report No. PNNL-SA-33642, Pacific Nortwest National Laboratory,
Washington DC, http://www.ntis.gov, 2001.
Mpanje, D., Gibbons, P., McDermott, R.: Social capital in vulnerable urban settings: An
analytical framework. Journal of International Humanitarian Action, 3, 4,
https://doi.org/10.1186/s41018-018-0032-9, 2018.
O' Brien, G., P. O' Keefe, H. Meena, J. Rose, and L. Wilson.: Climate adaptation
from a poverty perspective, Climate Policy, 8, 194-201,
https://doi.org/10.3763/cpol.2007.0430, 2008.
Parris, T., and Kates, R.: Characterizing and measuring sustainable development, Annual
Review of Environment and Resources, 28, 559-586,
https://doi.org/10.1146/annurev.energy.28.050302.105551, 2003.
Pelling M.: The vulnerability of cities: natural disasters and social resilience, London, Earthscan
Publications Ltd, the United Kingdom, ISBN 1-85383-830-6, 2003.
Perrow C. Disasters ever more? Reducing U.S. Vulnerabilities. In: Havidán Rodríguez, Enrico L.
Quarantelli and Russell R. Dynes (Ed.), Handbook of Disaster Research. (pp. 113-129).
Springer, New York, 2007.
Ramprasad V.: Debt and vulnerability: indebtedness, institutions and smallholder agriculture in
South India, The Journal of Peasant Studies, 46, 1286-1307,
https://doi.org/10.1080/03066150.2018.1460597, 2019.
Rygel L., David O. Sullivan and Yarnal B.: A method for constructing a social vulnerability index:
An application to hurricane storm surges in a developed country, Mitigation and Adaptation
Strategies for Global Change,11,741-764, http://doi.org/10.1007/s11027-006-0265-6, 2006.


Sagar Ambuj D., Najam A.: The human development index: a critical review, Ecological

2       Economics, 25, 249-264, https://doi.org/10.1016/S0921-8009(97)00168-7, 1998.

Sharma K.: The pandemic: Mirroring our fragilities,

4       https://en.unesco.org/courier/2020-3/pandemic-mirroring-our-fragilities, (last access: 3 May

5       2021), 2020.

SOPAC: The Environmental Vulnerability Index,

7       http://gsd.spc.int/sopac/evi/Files/EVI%202004%20Technical%20Report.pdf,    (last access:

8       10 March 2021), 2004.

Teng W.X., Xia J.W., Wan B.L.: On Rainstorm Vulnerability Assessment of Urban
Community: A Case Study on Yangpu District in Shanghai, Journal of Guangzhou
University (Social Science Edition), 17, 20-26, 60, 2018.
Timmerman, P.: Vulnerability, resilience and collapse of society, Toronto, Institute of
Environmental Studies, Canada, 1981.
Tunner B. L., Kasperson R. E., Matson P. A.: A Framework for vulnerability analysis in
sustainability science, Proceedings of the National Academy of Sciences of the United States
of America, 100, 8074-8079, https://doi.org/10.1073/pnas.1231335100, 2003.
Villa, F., McLEOD, H.: Environmental Vulnerability Indicators for Environmental Planning and
Decision-Making: Guidelines and Applications. Environmental Management, 29, 335-348,
https://doi.org/10.1007/s00267-001-0030-2, 2002.
Vincent, K.: Creating an index of social vulnerability to climate change for Africa,
http://www.nrel.colostate.edu/ftp/conant/SLM-knowledge_base/Vincent_2004.pdf,(last
access: 19 October 2021), 2004.
Weis Shawn W. Margles, Vera N. Agostini, Lynnette M. Roth, Ben Gilmer, Steven R. Schill, John
English Knowles, Ruth Blyther.: Assessing vulnerability: an integrated approach for mapping
adaptive capacity, sensitivity, and exposure, Climatic Change, 136,615-629,
https://doi.org/10.1007/s10584-016-1642-0, 2016.
Wisner, B., Blaikie P., Cannon T. and Davis I.: At Risk: Natural Hazards, People's Vulnerability
and Disasters, London, Routledge, the United Kingdom, ISBN 9780415084772, 2004.
Xu, J. and Takahashi M.: Progressing vulnerability of the immigrants in an urbanizing
village in coastal China, *Environment, Development and Sustainability, 23, 8*012-8026,
https://doi.org/10.1007/s10668-020-00914-8, 2021.
Xu, J. Li S.Z. Wu Z. Liu W.: The Vulnerability Assessment of Family Support for the Elderly
in Rural China: An Empirical Study Based on Data from Anhui, Population Research, 43,
91-101, https://rkyj.ruc.edu.cn/CN/Y2019/V43/I1/91, 2019.
Yang J.H.: Research on the Social Integration of China's Floating Population, Chinese Social
Sciences, 2015(2), 2015.
You W.J. and Zhang Y.L.: Research on Index System of Social Vulnerability for Flood Hazard,
Journal of Catastrophology, 28, 215-220, 2013.



Zhang Y.L. and You W.J.: Assessment of Social Vulnerability to Natural Disasters of Cities
2        Based on TOPSIS: A Case Study of Shanghai City, Journal of Catastrophology, 29, 109-114,
3        2014.

