# Peer review of "Identifying Vulnerable Population in the Urban"

_Natural Hazards and Earth System Sciences, 2022_

## Community Comment (CC1)

Comment on nhess-2022-277

1. *Table 2*

In the fifth column, some indicators have only one correlation, and some have two. The readers would be better understood if the authors identified positive and negative correlations for all indicators.

2. *3.3 data collection and analysis*

The analysis is based on a questionnaire. There is a lack of a table showing which specific questions make up the variables.

---

## Author Comment (AC1)

Dear Ms. Guochun Wu,

We highly appreciate your kind comments and suggestions to our manuscript. Your comments indeed make us have a deeper understanding on the subject of the paper, and the manuscript has been carefully revised according to your comments.

The answers for the questions and comments are as follows.

1. Table 2
In the fifth column, some indicators have only one correlation, and some have two. The readers would be better understood if the authors identified positive and negative correlations for all indicators.

Authors' responses:

Thanks for your valuable comments. We've made two revisions in Table 2 according to your suggestion, that's highlighted in red.

**Table 2** The Evaluation Index of Social Vulnerability

| Index | Indicator | Description | Source | Positive correlation (+) or negative correlation (-) to vulnerability |
|---|---|---|---|---|
| Exposure | Geographical location | Proximity to dangerous areas such as steep slope, riverbank, sea-shore, etc. | Pelling 2003, Moss et al. 2001. | Geographical location (+) |
| | Building | Flimsy constructions unable to withstand hazard impacts. | Wisner et al. 2004 | Building fragility (+) |
| | Public infrastructure | Unavailability of critical public infrastructure. | Moss et al. 2001, Cutter et al. 2003, Vincent 2004 | Access to public facilities (-) |
| Sensitivity | Health/physical ability | Physical ability of an individual or a group of people to withstand hazard impacts. | McCarthy et al. 2001, Pelling 2003, Moss et al. 2001, Hahn et al. 2009 | Bad physical condition (+) Good physical condition (-) |
| | Livelihood stability | Unstable livelihoods not conducive to increasing income, easily leading to poverty. | Marshall et al. 2007 | Unstable livelihood (+) |
| | Debt | Ways of life beyond mere subsistence level and lacks of long-term | Ramprasad 2019 | Debt (+) |

| | | | | |
|---|---|---|---|---|
| | | investment in disaster reduction. | | |
| | Renters | Lacks of access to costly housings and of sufficient shelter options. | Cutter et al. 2003 | Renters (+) |
| Adaptive capacity | Social inclusion | No participation in local decision-making leading to social marginalization concerning social identity, self-identification, rights, opportunities, participation, etc. | Yang 2015 | Social inclusion (-) |
| | Education | Ability to understand warning information and access to recovery information. | Cutter et al. 2003, Coulibaly et al. 2015 | Low education (+) High education (-) |
| | Family structure | A large number of people under the age of 18 and over 65 depending on more energy and resources to adapt to disasters. | Vincent 2004 Hahn et al. 2009, Coulibaly et al. 2015 | With the family member under the age of 18 and/or over 65 (+) Without the family member under the age of 18 and/or over 65 (-) |
| | Social capital | Access to information and resources, building trust and cohesion to reduce vulnerability. | Mpanje et al. 2018, Hahn et al. 2009 | Social capital (-) |
| | Social insurance | Normal hedge against losses caused by risks, lacking the ability to overcome adverse effects. | Burton et al. 1993, McCarthy et al. 2001, IPCC 2014 | Social security (-) |
| | Social security | Sufficient social welfare to improve living conditions, thereby enhancing disaster resilience, for example pensions or allowance increasing future expectations for the younger and guarantee subsistence of the elderly. | Vincent 2004, Wisner et al. 2004, Adger and Vincent 2005 | Social welfare (-) |
| | Disaster awareness | Lack of disaster awareness and experience which may impair the basic skills needed to protect oneself. | Wisner et al. 2004 | Awareness of disaster (-) |
| | Disaster preparedness | Inadequate disaster preparedness, for example food, water, rope etc., to reduce the ability to respond to disasters. | Wisner et al. 2004 | Disaster preparedness (-) |

2. 3.3 data collection and analysis
The analysis is based on a questionnaire. There is a lack of a table showing which specific questions make up the variables.

Authors' responses:

Thanks for your valuable comments. Given there are many tables in this paper, no new tables have been added. The questions of displaying each variable have been inserted into Table 4 and mentioned in the article. And the revised parts are marked in red.

In June and July of 2021, the preliminary interviews and the questionnaire surveys were conducted, respectively. First, we designed the questionnaires using the social vulnerability index (see Table 4) and the preliminary interviews with local residents.....

**Table 4** The determined and normalized variables

| Serial number | Variable | Description of Questions | Max | Min | Mean value | SD |
|---|---|---|---|---|---|---|
| 1 | Geographical location | Respondent's perception of the safety of his/her living place | 1 | 0 | 0.4372 | 0.1982 |
| 2 | Building | Respondent's evaluation of the safety of his/her housing | 1 | 0 | 0.4265 | 0.2103 |
| 3 | Critical infrastructure | a. Respondent's evaluation of the complete of his/her surrounding disaster prevention facilities (shelters, drainage facilities, embankments) b. Respondent's evaluation of the convenience of his/her surrounding facilities | 1 | 0 | 0.5245 | 0.2063 |
| 4 | Health/ Physical ability | Respondent's perception of his/her physical condition | 1 | 0 | 0.2872 | 0.2594 |
| 5 | Livelihood stability | Respondent's perception of the stability of his/her occupation (income) | 1 | 0 | 0.3863 | 0.2852 |
| 6 | Debt | Respondent whether he/she has loans | 1 | 0 | 0.1957 | 0.5076 |
| 7 | Renters | Respondent whether he/she owns or rents the house | 1 | 0 | 0.4599 | 0.5402 |
| 8 | Social inclusion | Respondent's perception of integration into local society | 1 | 0 | 0.2772 | 0.1788 |
| 9 | Education | Respondent's education level | 1 | 0 | 0.6064 | 0.2819 |
| 10 | Family structure | In the respondent's family, the proportion of children to be supported and the elderly to the total family population | 1 | 0 | 0.3871 | 0.2877 |
| 11 | Social capital | a. Respondent's evaluation about whether quickly get help from his/her family, relatives | 1 | 0 | 0.4526 | 0.2078 |

| | | | | | | |
|---|---|---|---|---|---|---|
| 12 | Social insurance | or friends after he/she has suffered disaster losses
b. Respondent's evaluation about whether quickly get help from the community, government or NGOs after he/she suffers from disaster losses
Respondent's evaluation of the sufficient of his/her insurance (such as personal safety insurance, housing insurance, other family property insurance, etc.) | 1 | 0 | 0.6614 | 0.3023 |
| 13 | Social security | Respondent's evaluation of the sufficient of his/her social security (such as medical security, pension, etc.) | 1 | 0 | 0.4603 | 0.2578 |
| 14 | Disaster awareness | a. Respondent's evaluation of his/her disaster knowledge and experience
b. Respondent's awareness about disasters in their living place | 1 | 0 | 0.5004 | 0.1647 |
| 15 | Disaster preparedness | a. Respondent's preparedness for disaster prevention and escape
b. Respondent's experience about participated in disaster drills | 1 | 0 | 0.7051 | 0.2973 |

---

## Author Comment (AC2)

**Response to RC1 Comments**

Dear reviewer,

We highly appreciate your valuable comments and suggestions. It has greatly improved the quality of our manuscript.We have made revisions one by one according to your comments and suggestions.

The answers for the suggestions and comments are as follows.

Q1: If possible, indicate the four community types (Type 1 - Type IV) in the Wuhan and Hongshan District map (Figure 2)

Authors' responses:

Thanks for your valuable suggestions. According to your suggestion. We've indicated the 11 communities in the Hongshan District map (Fig. 2), as follows.

[Figure]

**Figure 2.** Geographical features and administrative boundaries of Wuhan City and Hongshan District. The points of A-K show the locations of the communities where the questionnaire surveys were conducted.

Q2: In the methods section, the authors should clarify how the data were "weighting" in this context. For example, are you weighting the survey data to be representative of the target community and for non-response bias or are you applying a weighting algorithm to give more importance to certain measures of vulnerability?

Q3: Section 3.2 (Determination of weight): This section needs much more detail. Describe how you selected experts, the response rate, demographics of experts, methodology for eliciting ratings including the modality (in person, by phone/email, etc.), what prompts did you use, what scale did you

provide, how did you define each of the 15 indicators for the experts, etc. If the AHP method you used followed a standard procedure, please at least include a citation. Define what first-level and second-level index means. Describe how you "check the consistency of the judgment matrix". Include an interpretation of CR, CI, and RI for the reader.

Authors' responses:

Thank you very much for your comments. Firstly, we will explain the weighting part of the methodology. The empowerment of this study mainly adopts a combination of expert scoring method and Analytic Hierarchy Process. The calculation of the weight of this method is not based on research data, but on the scores given by the experts, that is different from methods such as principal component analysis and entropy method. After obtaining the weights of each indicator, combined with research data, the vulnerability of residents is calculated. Therefore, the purpose of weighting is to calculate vulnerability. Each weighting method has its advantages and disadvantages, and we have explained the reasons for choosing this method in the manuscript.

For consistency testing, generally, when the consistency ratio $CR<0.1$, it is considered that the consistency of the judgment matrix is acceptable, otherwise it needs to be corrected. 0.1 is the best solution obtained by the original author (Saaty 1980) through multiple Monte Carlo simulations. As this step is a necessary step in the Analytic Hierarchy Process, we have added references according to your suggestions when making revisions to facilitate better understanding by readers.

For the expert scoring method, the explanation in the manuscript is indeed not detailed enough, and we have made modifications: By snowball sampling, we firstly invited ten experts who are out of our research group from three countries (China, Japan and Indonesia) through email, including local people with disaster experience, local scholars with disaster experience, and/or researchers on related issues in sociology and geographers. By sending Table 2 (including explanations for each indicator) in a Word file, and specifying the steps for scoring 15 variables related to social vulnerability according to the degree of importance (Very important=5, More important=4, General important=3, Less important=2, Not important=1), we got feedback via email from all experts. There are no other prompts, and the expert response rate is 100%.

Q4: Page 9, lines 27-31: This main idea of paragraph is ambiguous. Are the authors arguing that migrants are de-stabilizing the adaptive capacity of a community or are the authors highlighting the reduced adaptive capacity of migrant individuals?

Authors' responses:

Thank you very much for your comments. For page 9, lines 27-31 in the original manuscript, after our discussion, we think that it is not very helpful in explaining the content of this section, so we have deleted it.

Q5: Section 3.3. (Data collection and analysis): This section is lacking in important details needed to evaluate the quality of the data generated from the survey. The authors should describe how they

determined a minimum sample size, how they constructed a sampling frame, and if they stratified communities based on demographic characteristics (e.g., migrant status). The authors should also include a description of the survey questions used to assess social vulnerability factors, how the survey was administered to respondents, and the refusal/non-response rate? The authors do not indicate that the survey data was weighting to account for demographic differences and/or non-response. The lack of weighting seriously undermines the generalizability and validity of the survey data. Without any indication that the data collected are representative of the underlying community, it is inadvisable to extrapolate the results beyond the sample of individuals included.

Authors' responses:

Thanks for your valuable suggestions.

About data collection, we have supplemented it based on your suggestion. Selecting the sampling method, it was taken into account that many urban migrants, especially low-skilled and low-secured representatives of migrant workers, were not fully included in the official urban population list. Therefore, we adopted the method of quota sampling to determine the sample size of each community based on the official data, and the preliminary research and interview data. Then, the required quantity for each community is determined in advance through mutual control quota analysis of the age, gender and household registration characteristics of the surveyed samples, and then distributed face-to-face until the target quantity is collected (Please see the pictures below).

The classification of the four types of communities is based on the official records, socio-economic data, the landscape of the community, and the determination of the number of different types of communities is also based on the preliminary research information. However, due to some communities not being allowed to enter during the research process, a total of 599 questionnaires were obtained from 11 communities.

[Figure]

[Figure]

[Figure]

Q6: Section 3.2 (lines 24-26, Calculation of Final Weight): This section requires much more detail. I don't understand what the authors are referring to in Step 2 (final weight) if Step 3 describes how they calculated the final weight.

Authors' responses:

Thank you very much for your comments. We are sorry for our mistake. In fact, steps 2 and 3 are one step, just in order to get more scientific results, we take the Arithmetic average, Geometric average, and Eigenvalue to calculate the weights, and then regard the average as the final weight of each indicator. We have made revisions in the manuscript.

[revised manuscript text omitted]

Q9: Table 5: It is more informative to show the percent of individuals in each community type that were high/mid/low vulnerability than the percent of individuals in each vulnerability category that lived in each community. For example, you show that 61% of high-vulnerability individuals lived in Type IV communities. However, only 27.6% of individuals who lived in Type IV communities are classified as high-vulnerability.

Q10: Page 17, lines 5-6: The authors state that, "The disparity in social vulnerability among inhabitants in various neighborhoods implies "residential segregation" in the metropolitan environments". However, their previous statement appears to contradict this conclusion: "A previous study (Turner et al. 2003) found that not only do social vulnerabilities vary between societies, communities, and groups, but also among residents in the same area/community. We have verified that using quantitative analysis receives similar findings (see Figure 5)."

Authors' responses:

Thank you very much for your comments. The main purpose of Table 5 is to display the distribution of populations with different vulnerabilities (high, medium, and low) among the four communities, rather than to clarify the distribution of populations with high, medium, and low vulnerabilities in the same type of community. This can also support the phenomenon of residential segregation mentioned in lines 17 and 5-6. We want to compare the vulnerability of residents between different communities rather than those within the same type of community. In addition, the research results presented in 4.1 and 4.2 are from different perspectives. Figure 5 shows social vulnerability between societies, communities, and groups, but also among residents in the same area/community. This is similar to previous research results, so we mentioned the study by Turner et al. 2003.

Q11: Page 19, lines 20-21: I believe that the authors are implying that occupation, household registration, gender and debt cannot reflect the variations in individual social vulnerability because there are no natural quantitative hierarchies to these factors. If so, that argument should be made more explicit. However, I would argue that it would be informative to show the proportion of individuals within each vulnerability group that belong to a specific vulnerable group (e.g., percent of workers employed in low-skill occupations, percent of individuals without household registration, etc.).

Authors' responses:

Thank you very much for your valuable suggestions. For Page 19, lines 20-21, we apologize for the unclear statement. What we actually want to express is that occupation, residence registration, gender and debt are categorical variables. Different from the continuous variables such as age and education in Table 6, their values cannot reflect individual vulnerability, so they cannot be put in Table 6. Then, we use Figure 7 to show the relationship between occupation, registered residence and vulnerability. In addition, the data results do not reflect the correlation between gender, debt and vulnerability, and it was not shown in the manuscript.

Q12: Page 20, lines 22-23: It is unclear the conclusion the authors present ("Although there are also some low-vulnerability individuals with the rural household registration, it can be argued that they are mainly engaged in state-owned enterprises, including public service units"). is supported by the survey data or if this is a hypothesis extrapolated by the authors.

Authors' responses:

Thank you very much for your comments. We found the results through data analysis of their occupational types, and we have made modifications to this sentence, as followed:
Although there are also some low-vulnerability individuals with the rural household registration, by analyzing their occupational types, it can be found that they are mainly engaged in state-owned enterprises, including public service units.

Q13: Page 16, line 20: The author indicate that communities of Type III have fewer scores than those of Type IV in terms of exposure and adaptive capacity, higher in sensitivity. I believe that they might have meant lower rather than fewer scores.

Authors' responses:

Thank you very much for your suggestion. Page 16, line 20 has been modified to lower scores.

For the language expression issue you mentioned, as other reviewers did not provide the same suggestions and due to time constraints, we have not made any grammar modifications this time. If you still think it is necessary, we will seek professional grammar editing services for correction in the next step.

---

## Author Comment (AC4)

**Response to RC2 Comments**

Dear reviewer,

We highly appreciate your valuable comments and suggestions. It has greatly improved the quality of our manuscript. We have made revisions one by one according to your comments and suggestions.

The answers for the suggestions and comments are as follows.

Q1: Line 15: Vulnerability is a key concept for both disaster risk and climate change adaptation. By analyzing the potential factors causing losses, it is possible to predict the extent to which a disaster will impact society in the future (Vincent 2004). The author mentions "factors contributing to losses"; are they referring to the concept of "root causes of a disaster"? Further clarification of this matter is required.

Authors' responses:

Thank you very much for your comments.
This study suggests that the causes of disasters and disaster losses come from both natural hazards and social conditions/social factors. The degree of damage caused by disasters is influenced by factors such as the exposure, sensitivity, and resilience of the social system to hazards. Certain social groups in some circumstances are prone to be impact towards hazards. Therefore, the factors that cause disaster losses we mentioned are not primarily focused on the hazard itself, but rather on the potential socio-factors that may cause damage in the hazard environment.

We have revised in the introduction section as follows:
Warming has become a predominant feature of the Earth's climate system, which has brought about changes in precipitation patterns and have increased in frequency of extreme weather such as heatwaves, droughts, forest fires, heavy rains, and floods. In recent years, these extreme weathers are continual to impact the vulnerable sections of society, bringing severe disaster losses around worldwide. By analyzing the potential socio-factors causing losses, it is possible to predict an extent to which a disaster will impact the society in the future (Vincent 2004). In order to reduce disaster losses and to improve disaster prevention capabilities, from the 1960s onward, vulnerability has formed an important research topic such as in the International Biological Program (IBP), the International Geosphere-Biosphere Programme (IGBP), the International Human Dimensions Programme on Global Environmental Change (IHDP), and the Intergovernmental Panel on Climate Change (IPCC) and so on (Zhang et al. 2008).
In urban areas, the emergence of social vulnerability is mainly determined by the instability of local society. Especially in the context of rapid urbanization, the continuous increase in population mobility poses severe challenges to local infrastructure, environments, and social structures. Socio-economic inequalities among the inhabitants are represented as a "mosaic" in the geographical space as a result of urban transformation. In addition to social-spatial isolation, such "mosaic" also leads to a redistribution of risk. Many studies on extreme events show that disastrous consequences do not only depend on a

hazard risk itself, but also are closely related to physical environments, social structures, and demographic characteristics of a geographic place (Perrow 2007; Bolin 2007). If one place is physically exposed to hazard risk, it will impact the population who live here in uneven ways (Huang et al. 2020). Although urban population mobility itself does not lead to vulnerability (Donner and Rodriguez 2008), the population are marginalized when the market and/or the government cannot provide adequate employment, water and sanitation facilities, housing, and medical services.

As the result of population dynamics and diverse demands for location, leading to the gradual decrease in the availability of safer lands, it is almost inevitable for human endeavors to be located in potentially dangerous places (Lavell 2003). For example, many migrants in Jakarta, Indonesia live in informal settlements called "Kampung" that are prone to flooding (Alzamil 2018). Ghana's capital Accra has 92 percent of migrants living in Old Fadama, a slum without tap-water or sanitation facilities (Awumbila 2014). The push to commercialize urban housing in China throughout the past 40 years of urbanization has widened disparities in living conditions. While existing old communities with poor living environments have not much improved, the living quality of new gated communities has significantly increased. This process has also created many marginal places, a hybrid of rural and urban systems characterized by high building density, unclear management rights and duties, and insufficient social infrastructure. The people who live there take the brunt of many urban disaster. Spatial and social differentiations in the city, resulting in the formation of new socially vulnerable groups based on various kinds of local community.

China is currently one of the most disaster-plagued countries in the world. There are many different kinds of disasters, and in recent years, their frequency, intensity, spatial scope, and duration has further expanded. As China is undergoing rapid urbanization, land expansion has created different types of community within and around the cities; the population, economy, and society are experiencing structural changes, making the society unstable. It is imperative to mitigate the impact of disasters on urban populations and communities, and case studies are expected to provide the policy bases for disaster risk reduction. The main purpose of this paper is to determine the degree of social vulnerability at the local level, and to identify the most vulnerable groups by focusing on the characteristics of social vulnerability within Chinese urban society from the micro perspective.

This paper mainly attempts to solve the following three questions:

 • What differences are in the vulnerability collectively for different types of urban communities?

 • What kinds of mosaic is seen in the urban areas? That is, how vulnerable populations are distributed across communities, and what are underlying reasons for this distribution?

 • Who are the most vulnerable groups in the city, and what characteristics do they have?

Q2: Line 34: Social vulnerability is influenced by various factors beyond social and economic status. There are as well as political conditions that affect an individual's or group's position and power in society and additionally, people's level of vulnerability may differ based on their life circumstances, age, and the time of year. Why the study did not consider the potential interactions between different social vulnerability indicators, which may affect the overall level of vulnerability?

Q3: Line 13: At the same time, the results also show that about 50% of urban registration holders are also at high and medium levels of social". Despite the quantitative results, did the author examine/documented urban social vulnerability from a more optimistic viewpoint, such as the innovative use of existing neighborhood groups for preparedness or the utilization of hazard and

vulnerability mapping? Additionally, did the author investigate cases of excellent coordination between municipalities and NGOs/CBOs regarding improvements in risk communication or increased sensitivity to the needs of population, both legal and illegal?

Authors' responses:

Thank you very much for your comments.

The main purpose of this paper is to determine the degree of social vulnerability at the local level, and to identify the most vulnerable groups by focusing on the characteristics of social vulnerability within Chinese urban society from the micro perspective. Our research refers to the Hazards of Place Model of vulnerability (as developed by Cutter, 1996) in the USA context, and applied the model to identify the vulnerability of persons living in risk zones. Vulnerability is conceived of in this model as both the biophysical and the social, but within a specified geographic domain. The HOP model integrates prospective exposures and societal resilience with a special focus on specific locations or areas (Kasperson et al. 1995; Cutter et al. 2000). It emphasizes that hazards should be the product of a specific region operating at the level of natural and social structures, and the vulnerability of a specific society to hazards. In terms of model reference and indicator selection, subjectivity is inevitable, which is one of the limitations of this manuscript.

Q4: Why did the study not consider the potential role of cultural and social factors in shaping social vulnerability and disaster risk?
Q5: Did the author recognize any limitations of this study? If so, it may be advantageous to incorporate these limitations in the manuscript.

Authors' responses:

Thank you very much for your valuable suggestions.

The current research does have limitations, and we apologize for not emphasizing them before. We have made a statement in the conclusion section.

The current research provides collective vulnerability of community. It compares the differences in vulnerability between different communities. However, the community referred was limited to administrative institutions with Chinese characteristics (*Shequ*). Although it also includes geographical and social meanings to some extent, in the Chinese context it is more inclined to the administrative dominion. Therefore, the discussion is mainly considered according to the administrative jurisdiction and does not involve the discussion of social networks, or social capital. The second limitation is in indicator selection and weight determination. The selection of different indicators and the adoption of different methods to calculate weights will produce different vulnerability results. Since there is still a lack of unified standards in the academic community, this study, although the selection is based on previous studies, still cannot avoid adding some subjective judgments. Future studies should explore suitable methods for determining indicators and weights.

Technical corrections: Given the dynamic nature of vulnerability, it would be advantageous to delineate a timeline that specifically identifies periods of heightened vulnerability over the course of the year,

particularly in relation to the influence of hazards. Such an approach would enable a more comprehensive appreciation of the "mosaic" of vulnerability within the research site.

Authors' responses:

Thank you very much for your comments. We strongly agree that the vulnerability you proposed is dynamic, but this dynamic nature is difficult to measure using quantitative methods, especially the quantitative methods used in this study. We conducted a questionnaire survey in June and July 2021. Although summer is indeed the peak period for disasters in the studied area, the questionnaire did not require respondents to only answer the situation during this time period, making it difficult to conduct dynamic vulnerability analysis on the timeline. We believe that the impact of different time periods on residents' vulnerability may have a greater impact on the exposure dimension. Your suggestion has indeed provided great insights, and we will continue paying attention to the temporal variation patterns of disaster occurrence time, frequency, intensity, and vulnerability in future research.

---

## Author Comment (AC5)

**Response to CC3 Comments**

Dear Dr. Song,

We highly appreciate your valuable comments and suggestions. It has greatly improved the quality of our manuscript.We have made revisions one by one according to your comments and suggestions.

The answers for the suggestions and comments are as follows.

Q1: The title, e.g., vulnerable population in the urban society: a case study in Wuhan, China, covers too much scope, which should be revised. This is because there are lots of kinds of disasters, as the authors mentioned. I suggest using the title: Identifying the vulnerable population in the urban society with flood disasters: A case study in Hongshan district of Wuhan, China.

Authors' responses:

Thanks for your valuable suggestions.

We have revised the title to: "Identifying the vulnerable population in the urban society: A case study in a flood-prone district of Wuhan, China", not mentioning the place name of Hongshan that is not so famous worldwide, but giving the place's feature to show a flood as a focal hazard.

Q2: In the abstract, the authors describe the findings of this investigation. Most finding of this investigation in the abstract can be figured out or guessed, even without conducting this investigation. Thus, it may be helpful to present some results quantitatively. For example, presenting central data in Table 5 in the abstract section is useful.

Authors' responses:

Thanks for your valuable suggestions.

Based on your suggestion, we have made revisions to the abstract. The results show the close interrelationships between different types of communities in terms of physical and built environments, and different levels of social vulnerability to disasters. The group of high vulnerability accounts for 12.9 percent of the 599 samples investigated, the group of medium vulnerability for 48.4 percent, and the group of low vulnerability for 38.7 percent. The higher vulnerability groups have the characteristics of low education, poor health, low annual income, unstable work, and insufficient social security. The quantitative understanding of the dissimilarity in the degree of social vulnerability between different communities and populations is of great significance for the reduction of social vulnerability and disaster risk specifically and pointedly.

Q3: Figures should be improved. For example, I failed to understand the meaning of Type I-4 in Figures 4-6 because the legend shows only limited information. Following stand-alone principles, each figure and table should be fully understood by the readership without referring to the main text.

Authors' responses:

Thank you very much for your valuable suggestions. We have added explanations to Figures 4-6 to help readers better understand the information the figures are trying to show, as follows:

[Figure]

**Figure 4.** Social Vulnerability Box Plot of 4 type communities. The boxplot in is used to represent the central location and distribution range of vulnerability data for the four types of communities, and to compare them. The four colors represented in the legend represent four different community types, each consisting of multiple communities (see Table 1). There is a line in the middle of the box, representing the median of the data; The top and bottom of the box are respectively the upper quartile (Q3) and the lower quartile (Q1) of the data; The top and bottom lines represent the maximum and minimum values of the group of data, respectively. Some points distributed outside represent outlier in the data. This figure can not only show the distribution, outlier, fluctuation and stability of each type of community vulnerability, but also compare the difference of distribution and value of different types of community vulnerability. *Note*: $p < .01$*** (= .000)

[Figure]

**Figure 5.** Exposure, sensitivity, and adaptive capacity of four types community. The bubble chart shows three variables (exposure, sensitivity, and adaptability) for four types of communities. Exposure and sensitivity correspond to values on the *X*-axis and *Y*-axis, respectively, and adaptability is represented by the size of the bubble. The four different colors in the legend represent four types of communities, and the dot size is used to explain the size of adaptability. Through Figure 5, not only can the overall exposure, sensitivity, and adaptability of the study area be displayed, but also the differences in exposure, sensitivity, and adaptability of different types of communities can be compared.

[Figure]

**Figure 6.** The distribution and characteristics of high, medium and low-level vulnerability. The figure

horizontally represents the distribution of high, medium, and low vulnerability populations in the four types of communities. Vertically, a) Range value is the nuclear density curve of the vulnerable population, with a higher peak indicating a more concentrated level of vulnerability (with smaller differences in vulnerability). Conversely, a lower peak indicating a more dispersed level of vulnerability (with larger differences in vulnerability). At the same time, the concentration range of its vulnerability values can be determined; b) Exposure-Sensitivity represents the correlation between the exposure and sensitivity of vulnerable populations in the four types of communities, with the *X*-axis indicating exposure and the *Y*-axis indicating sensitivity; c) Exposure-Adaptive Capacity represents the correlation between the exposure and adaptability of highly vulnerable populations in the four types of communities, with the *X*-axis indicating exposure and the *Y*-axis indicating adaptability; d) Sensitivity-Adaptive capacity represents the correlation between sensitivity and adaptability of vulnerable populations in the four types of communities, with the *X*-axis indicating sensitivity and the *Y*-axis indicating adaptability.

---

## Author Response (AR2)

**Response to Anonymous Referee #2**

Dear reviewer,

We are extremely grateful for your constructive feedback. It has greatly improved the quality of our manuscript.

We have made revisions one by one according to your comments and suggestions, and the revised manuscript has been uploaded together with the response letter. In the revised manuscript, the newly added and altered sections are highlighted in red, to hopefully facilitate your review.

The answers for the suggestions and comments are as follows.

Q: Regarding line 17 on page 14, I would appreciate it if you could clarify the basis on which the author divided the communities into four categories. Was there any previous study used as a foundation for this classification?

Authors' responses:

Thank you for your insightful suggestions and ongoing support.

In our study, we outline the rationale for classifying the target community into four types. Table 1 provides a description of the characteristics of these community types: Within China's metropolitan regions, the housing reform policy has spurred a socioeconomic spatial division within communities. Analyzing Wuhan City's district housing plan, we categorized target communities into four types (Table 1): high-grade residences (Type I), newly demolished and rebuilt communities (Type II), old demolished and reconstructed communities (Type III), and urban villages (Type IV). Urbanization and land expansion have furthered this spatial diversity, leading to varying stages of development across communities. Consequently, distinctions emerge in scenery, public facilities, and administrative management levels.

We apologize for any inconvenience caused by readability issues. We've revise the sentence to enhance clarity and conciseness. "Within the ambit of our study, eleven communities, labeled A to K, were systematically categorized into Types I to IV (refer to Section 2)".

**Response to Referee #3**

Dear reviewer,

We are extremely grateful for your constructive feedback. It has greatly improved the quality of our manuscript.

We have made revisions one by one according to your comments and suggestions, and the revised manuscript has been uploaded together with the response letter. In the revised manuscript, the newly added and altered sections are highlighted in red, to hopefully facilitate your review.

The answers for the suggestions and comments are as follows.

Q1: The main flood event for which the paper conducted the survey is the one from 1998, and we are in the year 2023, 25 years after. There is no age distribution of the ones who responded to the interview, to see for how many the 1998 flood was still relevant. Moreover, after 25 years people tend to forget. How was that taken into account while studying social vulnerability? were the city areas in 1998, the same as now, for sure not. if the community type changed, how did it affect their flood perception.

Authors' responses:

Thank you very much for your suggestions.

The context of this paper is not centered around the 1998 Yangtze River flood. In addition to the 1998 flood, the Wuhan region has experienced multiple instances of flood disasters, resulting in significant economic losses and casualties. Therefore, Wuhan is considered a flood-prone area. We discuss this in the paper:"Changes in lake water levels have had a weaker relationship with the Yangtze River since 2000, when the dam was completed. However,the main effects were precipitation and industrial, agricultural, and household water use. As a result, the flooding induced by the rising water level of the inner lakes was the primary hazard risk in Hongshan District."

In the "Data collection and analysis" section, we wrote conducting preliminary interviews and questionnaire surveys in June and July 2021. The inquiries during the investigation focused on the current situation, not their recollections of the events in 1998. We apologize for any inconvenience caused by readability issues.

Q2: The methodology section lacks detail. For example it is not very clear how from the three weights of the indicators for exposure, one would come up with a weight for exposure itself?

Authors' responses:

Thanks for your insightful suggestions.

And we apologize for any inconvenience caused by readability issues. In the "Determination of weight" section, we outline the method for calculating the weight of each index/indicator. "To enhance scientific rigor, we employed the arithmetic average, geometric average (Dvorák 2016), and eigenvalue (Golub and Van der Vorst 2000) methods. Subsequently, we considered the average derived from these calculations as the final weight for each indicator (refer to Table 3)." References have been included to illustrate the standardization process.

Q3: Indicators seem to have too many digits after the dot. For example how can a weight of 0.179836 have more influence than 0.179 or 0.18. Did the researchers try to see.

Authors' responses:

Thank you for your insightful suggestions.
Following your advice, we have made modifications to the indicators, retaining two digits after the dot.

Q4: Figures have too long captions, difficult to follow. Titles/captions should be short and suggestive and all the other explanations should be given in the main text of the manuscript.

Authors' responses:

Thank you very much for your suggestions.
For image captions, these suggestions were shared by the initial two reviewers. The original manuscript included only titles, lacking accompanying explanations. Ensuring reader comprehension of images without delving into the main text may be pivotal. If you maintain this stance, integrating them into the main text is a possibility. However, further discussions with additional reviewers or editors are necessary.

Q5: Conclusions need to be more critical about the results. In the end all is lummped in a number, the vulnerability as an index need to be used along with other data.

Authors' responses:

Thank you for your insightful suggestions.
In our last revision, we engaged professional editing services, refining the entire manuscript with the assistance of a native speaker. This time, we have thoroughly examined the entire text in line with your recommendations. And the revised parts have been highlighted in red.

Q6: Overall the paper is well written, though some parts need rewriting, for example page 11, line 2-9.

Authors' responses:

Thanks for your insightful suggestions.

In accordance with your advice, we've revised the conclusion, highlighting key research findings and presenting results that diverge from both previous studies and the prevailing notion.